# On the generalization of learning algorithms that do not converge

**Nisha Chandramoorthy**
Institute for Data, Systems and Society
Massachusetts Institute of Technology
nishac@mit.edu

**Andreas Loukas**
Prescient Design
Genentech, Roche
andreas.loukas@roche.com

**Khashayar Gatmiry**
Electrical Engineering and Computer Science
Massachusetts Institute of Technology
gatmiry@mit.edu

**Stefanie Jegelka**
Electrical Engineering and Computer Science
Massachusetts Institute of Technology
stefje@mit.edu

## Abstract

Generalization analyses of deep learning typically assume that the training converges to a fixed point. But, recent results indicate that in practice, the weights of deep neural networks optimized with stochastic gradient descent often oscillate indefinitely. To reduce this discrepancy between theory and practice, this paper focuses on the generalization of neural networks whose training dynamics do not necessarily converge to fixed points. Our main contribution is to propose a notion of *statistical algorithmic stability* (SAS) that extends classical algorithmic stability to non-convergent algorithms and to study its connection to generalization. This ergodic-theoretic approach leads to new insights when compared to the traditional optimization and learning theory perspectives. We prove that the stability of the time-asymptotic behavior of a learning algorithm relates to its generalization and empirically demonstrate how loss dynamics can provide clues to generalization performance. Our findings provide evidence that networks that "train stably generalize better" even when the training continues indefinitely and the weights do not converge.

## 1 Introduction

It is common practice that when the training loss of a neural networks converges close to some value, the learning algorithm—typically some variant of Stochastic Gradient Descent (SGD)—is terminated. Perhaps surprisingly, recent works indicate that the network function at termination time is typically not a fixed point of the learning algorithm: if run longer, the gradient norm does not vanish and the learning algorithm outputs functions with significantly different parameters [Cohen et al., 2021, Zhang et al., 2022, Lobacheva et al., 2021]. This observation stands in contrast to common generalization analyses that assume convergence of the training algorithm to a fixed point. It also raises the question of when (and why) non-convergent learning algorithms should be expected to generalize.

Standard approaches for obtaining generalization bounds find ways to bound the complexity measure of the function class expressible by a neural network in a data-independent e.g., [Vapnik, 1999, Bartlett et al., 2017, Neyshabur et al., 2018, Golowich et al., 2018, Bartlett et al., 2019, Arora et al., 2018] or data-dependent manner, e.g., [von Luxburg and Bousquet, 2004, Xu and Mannor, 2012, Sokolić et al., 2017]. Nevertheless, even with recent technical improvements, by and large these

36th Conference on Neural Information Processing Systems (NeurIPS 2022).

approaches do not explicitly account for the relationship between generalization and the dynamics of the learning algorithm.

A different approach to generalization analysis yields algorithm-dependent bounds by connecting the generalization performance of the trained network to the stability of the training to data perturbations [Bousquet and Elisseeff, 2002, Feldman and Vondrak, 2018, Kuzborskij and Lampert, 2018, Bousquet et al., 2020, Zhang et al., 2021b, Rakhlin, 2006]. A key claim of these works has been that networks that are trained fast generalize better. This claim, though intuitively meaningful, hinges on the premise of a convergent algorithm, which deviates from the observed behavior of deep neural network training. Further work has been done for settings where the algorithm provably converges to a fixed point, such as two-layer overparameterized networks, deep linear networks, certain matrix factorizations, etc. [Frei et al., 2019, Allen-Zhu et al., 2019, Soudry et al., 2018, Arora et al., 2019b, Gunasekar et al., 2018, Haeffele and Vidal, 2015]. In a more general setting, Hardt et al. [2016] prove algorithmic stability-based generalization bounds that worsen with increasing training time, whereas Loukas et al. [2021] connect stable training dynamics near convergence under Dropout with good generalization. For algorithms that do not converge (close) to a fixed point, the above analyses result in vacuous bounds.

**Contributions.** We start with the supposition that robustness of the learning algorithm's *fixed points* to small changes in the training set is *not* the mechanism underlying the good generalization properties of deep networks. This is because, as is commonly known and many recent works indirectly remark upon [Cohen et al., 2021, Ahn et al., 2022, Garipov et al., 2018, Zhang et al., 2022], training can be *linearly unstable*: even if assuming the presence of isolated local minima that the learning algorithm converges to, the weights encountered in training are not robust to small perturbations. Small perturbations can accumulate over time leading to completely different orbits in weight space. Yet, despite unstable dynamics, the generalization properties of two completely different fixed points (e.g., obtained from two different random initializations or due to a different order of SGD updates) are typically comparable.

Our first contribution entails proposing a generalization of stability that applies to non-converging algorithms. To achieve this, we depart from the typical optimization perspective of analyzing the generalization properties of minima [Keskar et al., 2016, Ge et al., 2015, Sagun et al., 2016]. Instead, we study the generalization performance of the learning algorithm *on average* (see section 2.2). Adopting a statistical viewpoint, in Section 3 we define a new notion of *statistical algorithmic stability* that measures the stability of this average performance to perturbations of the training data, as opposed to stability of individual orbits in the weight space. Section 3 then proves upper bounds for the average generalization error based on our new notion of statistical stability.

We then move on to consider how statistical stability connects to the behavior of the loss function. To this end, in Section 4 we provide conditions such that the spectral gap of a Markov operator associated with the dynamics of the loss function is predictive of statistical algorithmic stability and hence of generalization. Empirically, we estimate this spectral gap based on the rate of convergence of ergodic averages of the loss function. We show via experiments that our estimate correlates with generalization for image classification under corruption.

## 1.1 Related work

Our work builds on previous analyses of the stability of learning algorithms that converge [Bousquet and Elisseeff, 2002, Feldman and Vondrak, 2018, Kuzborskij and Lampert, 2018, Bousquet et al., 2020, Zhang et al., 2021b]. Below, we also briefly discuss connections (beyond the literature on stability) with previous analyses on the dynamics of learning algorithms.

**Convergence and generalization of SGD.** Exponential convergence rates to global minima are known for SGD in the smooth and convex setting, with decaying learning rates and small constant learning rates (e.g., [Moulines and Bach, 2011, Needell et al., 2014]). In the training of deep neural networks, which is the focus of this paper, the loss function is non-convex and is typically not well-approximated locally by convex functions [Ma et al., 2018]. However, in the overparameterized regime, a Polyak-Lojasiewicz-type inequality that is automatically satisfied when the loss is smooth and convex can be shown to hold in the non-convex setting as well [Liu et al., 2022, Ma et al., 2018], allowing convergence to a global minimum. In certain non-convex problems, convergence to local minima and even global minima has been proved, assuming a strict saddle property [Ge et al., 2015]

or local convexity [Kleinberg et al., 2018]. Raginsky et al. [2017] show a favorable convergence for a slightly different version of SGD by approximating the dynamics with a suitable continuous time Langevin diffusion. A number of previous works [Safran and Shamir, 2016, Freeman and Bruna, 2016, Li and Yuan, 2017, Nguyen et al., 2018] have also provided arguments that SGD converges to a solution that generalizes given suitable initialization and significant overparameterization. A few works have studied the nonlinear dynamics of training [Kong and Tao, 2020] and provided insights into generalization in the overparameterized regime [Dogra and Redman, 2020, Saxe et al., 2013, Advani et al., 2020]. Another line of work considers the mean field limit of SGD dynamics to show generalization [Mei et al., 2019, Gabrié et al., 2018, Chen et al., 2020]. The ergodic theoretic perspective we adopt here deviates from the optimization perspectives but relates to the mean field perspective in that we implicitly consider (see section 4) the evolution of probability distributions on weight space.

**Flat minima.** In the generalization literature, flat minima correspond to large connected regions with low empirical error in weight space. Flat minima have been argued to be related to the complexity of the resulting hypothesis and, hence, can imply good generalization Hochreiter and Schmidhuber [1997]. It has also been shown that SGD converges more frequently to flat minima when the batch size is small or the learning rate and the number of iterations are suitably adjusted [Keskar et al., 2016, Hoffer et al., 2017, Jastrzębski et al., 2017, Smith and Le, 2018, Zhang et al., 2018]. Some works consider the flat/sharp dichotomy an oversimplification [Dinh et al., 2017, Sagun et al., 2017, He et al., 2019], e.g., using a reparameterization argument [Sagun et al., 2017]. The eigenvalues of the hessian of the loss affect the local linear stability of optimization orbits. However, given that we adopt the time-asymptotic/statistical picture, we do not explicitly make assumptions on local stability (flatness/sharpness) or the topology of the loss landscape [Montanari and Zhong, 2020, Venturi et al., 2019].

**SGD as a Bayesian sampler.** The idea of looking at distributions of learners permeates PAC-Bayesian analyses of generalization [McAllester, 1999, Dziugaite and Roy, 2017, Zhou et al., 2019, Pitas, 2020]. In contrast to Bayesian posteriors of the parameters, we study parameter distributions generated by the learning algorithms. Previous empirical work has suggested that SGD operates almost like a Bayesian sampler [Mingard et al., 2021] but the connection remains to be fully understood.

## 2    Local descent algorithms as dynamical systems: a statistical viewpoint

This section lays out the basic definitions and assumptions of our work. We begin in Section 2.1 by introducing the learning problem and discussing learning algorithms from a dynamical systems perspective. Section 2.2 then puts forth the notions of statistical convergence that give rise to our main results.

### 2.1    Learning as a dynamical system

We consider the supervised learning setup: a learner is given a training set $S = \{z_1, \ldots, z_n\}$ consisting of $n$ pairs $z_i \equiv (x_i, y_i)$ of inputs $x_i \in \mathbb{R}^d$ and their corresponding labels $y_i \in \mathcal{Y} \subseteq \mathbb{R}$, drawn i.i.d. from a distribution $\mathcal{D}$. A class of parametric models or *hypotheses* $h : \mathbb{R}^d \times \mathbb{W} \to \mathbb{R}$ on a parameter space $\mathbb{W}$ gives possible input-to-output relationships, $h(w, \cdot) : \mathbb{R}^d \to \mathbb{R}$. The learner is given a risk or *loss* function $\ell : (\mathbb{R}^d \times \mathbb{R}) \times \mathbb{W} \to \mathbb{R}^+$ that describes the error of a given hypothesis. Common choices for loss functions are the mean-squared, hinge and cross-entropy loss. The learner attempts to minimize the population risk $R_S(h(\cdot, w)) = \mathbb{E}_{z \sim \mathcal{D}} \ell(z, w)$, by minimizing the empirical risk

$$\hat{R}_S(h(\cdot, w)) = L_S(w) = \frac{1}{n} \sum_{z_i \in S} \ell(z_i, w). \tag{1}$$

This minimization is achieved using a local descent algorithm given by iterative updates $\phi_S : \mathbb{W} \to \mathbb{W}$ of the form

$$w_{t+1} = \phi_S(w_t) := w_t - \eta_t \, \hat{\nabla} L_S(w_t), \tag{2}$$

where $\eta_t$ is the learning rate or step size at time $t$.

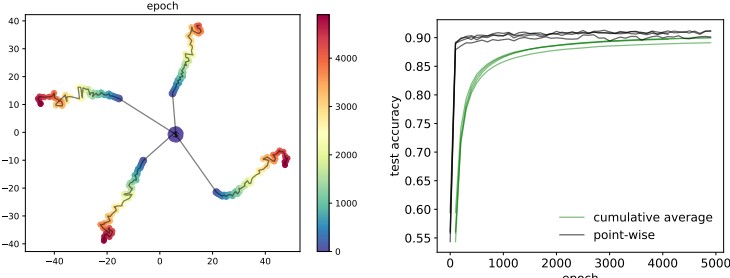

Figure 1: Although the training of neural networks depends on the initialization and does not necessarily converge in the weight space, certain functionals of the learned hypothesis can converge in distribution independently of initial conditions. Left: The orbits of the 2nd layer weights of four different VGG16 models trained on a CIFAR10 dataset using SGD with step size 0.01. We embed the orbits in the plane by performing PCA onto 40 dimensions followed by t-SNE. Colors indicate epoch. Right: Corresponding test accuracy and cumulative average values as a function of epoch.

In the gradient descent (GD) algorithm, $\hat{\nabla}L_S(w) = \nabla L_S(w)$ is the gradient w.r.t. $w$, and the iterates of $\phi_S$ represent a deterministic dynamical system. The critical points (including saddle points, local and global minima) of $L_S$ are fixed points of $\phi_S$ (i.e., points $w$ such that $\phi_S w = w$). For stochastic gradient descent (SGD), $\hat{\nabla}L_S(w)$ is a random variable given by the sample mean of gradients $\nabla\ell(\cdot, w)$ over a batch of samples in $S$. In this case, $\phi_S$ is a random dynamical system and critical points of $L_S$ are not necessarily fixed points of $\phi_S$.

To unify the definition of $\phi_S$ for both GD and SGD, we introduce the random variable $\Xi$ that indicates the choice of batch. Suppose the batch size $m$ is fixed: $m < n$ for SGD and $m = n$ for GD. Denote by $[n]$ the set $\{1, 2, \cdots, n\}$ and let $\Xi_t$ be a collection of $m$ elements from $[n]$ chosen uniformly at random. Then, we may write $\phi_S$ as

$$w_{t+1} = \phi_S(w_t) = w_t - \eta_t \hat{\nabla}L_S(w_t|\Xi_t) = w_t - \frac{\eta_t}{m}\sum_{i \in \Xi_t} \nabla\ell(z_i, w_t), \tag{3}$$

with $\Xi_t$ being the set $[n]$ at all $t$ for GD.

In our investigation, we opt for simplicity and consider a fixed learning rate $\eta_t = \eta$. More generally, our analysis can be extended to learning rates that asymptotically converge to $\eta$. A rapidly vanishing learning rate can obscure the learning dynamics by enforcing convergence to arbitrary points irrespective of the points' losses. In contrast, learning rates that do not decay to zero can induce interesting transitions between neighborhoods of critical points and the iterates $w_t$ are generally more exploratory of the loss landscape.

## 2.2 Statistical convergence of the learning algorithm

In deep learning, any specific choice of $w_T$, at some time $T$, is arbitrary as the loss landscape is generally non-convex and stochastic learning algorithms are the norm. Hence, rather than focusing on a single hypothesis, it is attractive to consider the average generalization error induced by the stochastic dynamical system $\phi_S$. We adopt this *statistical* perspective, rather than focus on a single orbit of $\phi_S$. Thus, the setting below applies to any asymptotic behavior of orbits of the (typically nonlinear) dynamics $\phi_S$ including fixed points, periodic, quasiperiodic and chaotic orbits.

To analyze the generalization performance without making assumptions on the global stability properties of individual orbits, we make an assumption about the ergodic properties of $\phi_S$ that reasonably matches empirical evidence. To describe the assumption, we give a short primer on the relevant concepts from ergodic theory of dynamical systems.

**Invariant measures.** Denote the Borel sigma algebra on $\mathbb{W}$ by $\mathcal{B}(\mathbb{W})$. A probability measure $\mu_S$ is called *invariant* for $\phi_S$ if $\mu_S(A) = \mu_S \circ \phi_S^{-1}(A)$ for all Borel sets $A \in \mathcal{B}(\mathbb{W})$. Intuitively, for any subset of the weight space, the probability that the dynamical system occupies it does not change. See Liverani [2004] for an introduction to invariant measures.

If $\phi_S$ is a deterministic continuous function on the set $\mathbb{W}$ and $\mathbb{W}$ is compact, classical ergodic theorems (see e.g., Theorem 4.1.1 of Katok and Hasselblatt [1997]) give the existence of at least one invariant, ergodic measure for $\phi_S$. Ergodic measures are a pertinent class of invariant measures in practice, that allow us to understand the statistical properties of a system through evolution of individual orbits. For an ergodic measure, sets that are invariant under the dynamics $\phi_S$ (such as a set of periodic points, quasiperiodic or chaotic attractors) are trivial: they have measure 0 or 1.

When $\phi_S$ is an SGD update, Dieuleveut et al. [2020] show the existence of invariant distributions for convex loss functions, whereas Fort and Pagès [1999] and Mattingly et al. [2002] show the existence of invariant distributions in more general settings under mild assumptions on the gradient noise (in the estimate $\hat{\nabla} L_S$). Several works have analyzed the asymptotic properties of such invariant distributions, e.g., Chee and Toulis [2018] analyze the oscillation of the iterates $w_t$ about a mean that is a critical point, as a *phase* separate from the transient phase of convergence to this invariant distribution. Kong and Tao [2020] study the effect of large learning rates (see also [Wang et al., 2021] for matrix factorization problems) and multiscale loss functions to show the existence of Gibbs invariant measures in GD.

**Ergodicity.** When an invariant measure $\mu_S$ is ergodic for $\phi_S$, time averages converge to ensemble averages according to $\mu_S$. That is, for $\mu_S$-almost every initial state $w_0$, and for all continuous scalar functions $h$:

$$\frac{1}{T} \sum_{t=0}^{T-1} h(w_t) \rightarrow \mathbb{E}_{w \sim \mu_S} h(w) \tag{4}$$

and the dependence on initial conditions $w_0$ is forgotten.

Unfortunately, we cannot generally expect the dynamics of the learning algorithm to have a unique ergodic invariant measure on $\mathbb{W}$. Instead, $\phi_S$-invariant sets could be smaller subsets of $\mathbb{W}$ with different ergodic invariant measures supported on them. As a result, starting from two different points chosen Lebesgue almost everywhere (or sampled from any probability density on $\mathbb{W}$), we do not expect time averages to converge to the same limit. Indeed, as shown in Figure 1, the weights visited by $\phi_S$ depend on the initial conditions and can vary significantly across different runs. In other words, the limit of infinite time averages described in (4) does depend on $w_0$.

**Dynamics of the hypothesis.** To circumvent the above issues, rather than focusing on the weights, we consider the dynamics of the *learned hypothesis* $h(\cdot, w_t)$. Specifically, suppose that the infinite-time averages of the learned hypothesis are ergodic: that is, the time averages of $h(z, w_t)$ converge to the same function irrespective of the initial function $h(z, w_0)$. Then, functionals that depend only on $h$ and not explicitly on the weights are also ergodic. This is one such scenario that leads to our main assumption: the time averages of loss functions are ergodic.

**Assumption 1.** *Given any $S \sim \mathcal{D}^n$, there exists a map $z \rightarrow \langle \ell_z \rangle_S$, such that for Lebesgue-a.e. $w_0$ and every $z \in \mathbb{R}^d \times \mathbb{R}$, the following holds:*

$$\lim_{T \to \infty} \frac{1}{T} \sum_{t=0}^{T-1} \ell(z, w_t) = \langle \ell_z \rangle_S \in \mathbb{R}^+, \tag{5}$$

*where $\{w_t = \phi_S(w_{t-1})\}$ is an orbit of $\phi_S$.*

As an intuitive justification for Assumption 1, symmetries may result in several sets of parameters (weights and biases) that represent the same network function $h$. Hence, assuming that there exists a unique probability on the space of neural network functions that is ergodic is less restrictive and closer to reality than assuming a unique ergodic measure on the parameter space. Furthermore, ergodicity of the network functions is a stronger condition than our Assumption 1 and serves to illustrate one possible sufficient condition for our assumption to hold.

Figure 1 provides evidence that Assumption 1 can hold in practice. It shows two observables calculated along 4 different orbits of a VGG16 model trained using SGD with momentum (batches of size 128, learning rate of 0.01, momentum of 0.9). On the left, we can see a low-dimensional projection of the second layer weights obtained by projecting the weights onto 40 dimensions using PCA and then employing t-SNE [Van der Maaten and Hinton, 2008], whereas the observable on the right is the test accuracy, which is a functional of the hypothesis function. Here, the test accuracy can

be seen to converge to a distribution independent of the initial conditions, whilst the weight orbits vary significantly across different initializations. This experiment corroborates our hypothesis that time averages of functionals on the loss space converge to distributions independent of the initial condition, even if time averages of a generic observable on the weight space do not. See Remark 1 on the idea that a larger learning rate induces ergodicity and Appendix A, which illustrates the idea with a toy example.

## 3 Statistical stability implies generalization

In Section 3.1, we generalize algorithmic stability beyond algorithms that converge to fixed points. Section 3.2 then proceeds to examine the implications of our definition to generalization error.

### 3.1 Statistical algorithmic stability

Classically, the derivations of algorithmic stability-based generalization (see e.g., Hardt et al. [2016], Chapter 14 of Mohri et al. [2018], Bousquet et al. [2020]) utilize an input perturbation of one element in $S$ by replacing it with a different element from the input distribution $\mathcal{D}$:

**Defnition 1** (Algorithmic stability, adapted from [Bousquet and Elisseeff, 2002]). *Consider the weights $w_S^*$ and $w_{S'}^*$ obtained by running the learning algorithm on two training sets $S$, $S'$ sampled from $\mathcal{D}$ that differ by exactly one sample. We say that the learning algorithm is algorithmically stable (AS) with a stability coefficient $\beta \geq 0$ if*

$$\beta = \sup \left\{ |\ell(z, w_S^*) - \ell(z, w_{S'}^*)| : z \in \mathbb{R}^d \times \mathbb{R} \right\}. \tag{6}$$

We refer to this type of perturbation as a *stochastic* perturbation. Algorithmic stability does not directly apply to learning algorithms that do not converge to a fixed point. Hence, next, we extend algorithmic stability to loss *statistics*. The resulting notion of *statistical algorithmic stability (SAS)* extends algorithmic stability to algorithms whose loss statistics converge as prescribed by Assumption 1, even if the weights do not converge.

**Defnition 2** (Statistical algorithmic stability). *We say that a learning algorithm with loss statistics denoted by $\langle \ell_z \rangle_S$ is statistically algorithmically stable (SAS) with a stability coefficient $\beta \geq 0$ if*

$$\beta = \sup \left\{ |\langle \ell_z \rangle_S - \langle \ell_z \rangle_{S'}| : z \in \mathbb{R}^d \times \mathbb{R} \right\}, \tag{7}$$

*where $S$, $S'$ differ in exactly one element.*

In the above definition, $\langle \ell_z \rangle_{S'}$ refers to the ergodic average of $\ell(z, \cdot)$ observed along almost every orbit of $\phi_{S'}$. A higher value of $\beta$ indicates a lower SAS algorithm. The definition of SAS differs from Definition 1 as we do not assume convergence of the learning algorithm. Nevertheless, when every orbit of $\phi_S$ converges to a fixed point $w_S^*$, for almost every $S$, then Definition (7) reduces to the standard notion of algorithmic stability since ergodic averages along almost every orbit converge to $\ell(z, w_S^*)$.

Crucially, and in line with the observations in Figure 1, we quantify stability based on the statistics of the *loss function* and not the loss function at an ensemble mean or at any one point in the weight space $\mathbb{W}$. In other words, the definition of SAS does not use or provide information about the algorithmic stability of $\mathbb{E}_{w \in \mu_S} w$ for any invariant measure $\mu_S$.

### 3.2 Learning theoretic implications

It is well-known that Definition 1 leads to generalization bounds (see [Hardt et al., 2016, Bousquet et al., 2020] and references therein). Next, we show that the more general statistical algorithmic stability from Definition 2 also can imply a notion of generalization.

To obtain generalization bounds, we first redefine the empirical and population risk to use the loss statistics rather than loss values at fixed points:

$$\hat{R}_S = \frac{1}{n} \sum_{z \in S} \langle \ell_z \rangle_S \quad \text{and} \quad R_S = \mathbb{E}_{z \sim D} \langle \ell_z \rangle_S. \tag{8}$$

Although the definitions above appear deceivingly similar to the standard ones introduced in Section 2.1, the two notions of risk are defined on different spaces, with the standard ones being maps from the *hypothesis space* $\mathcal{H}$ to scalar values and (8) being maps from the space of *learning algorithms*. That is, the latter risks only depend on algorithmic parameters and are not functions of $\mathcal{H}$. Using these definitions, the following generalization bound holds:

**Theorem 1.** *Given a $\beta$-statistically algorithmically stable algorithm $\phi_S$ (see Definition 2), for any $\delta \in (0, 1)$, with probability greater than $1 - \delta$,*

$$R_S \leq \hat{R}_S + \beta + 2 \left( n\beta + L \right) \sqrt{\frac{\log(2/\delta)}{2n}}, \tag{9}$$

*where $L := \sup_z \sup_{w \in \mathbb{W}} |\ell(z, w)|$ is an upper bound on the loss.*

The proof can be found in Appendix B and is based on applying Mcdiarmid's inequality to the generalization gap. Like Bousquet and Elisseeff [2002]'s bound, the SAS coefficient $\beta$ has to decay at least as fast as $\sim \mathcal{O}(1/\sqrt{n})$ for the generalization bound to be non-vacuous. Note that our proof is also naturally coupled with [Bousquet et al., 2020, Feldman and Vondrak, 2018]'s improved technique, which provides tighter bounds when $\beta \sim \mathcal{O}(1/\sqrt{n})$. (see Appendix B).

## 4 Dynamical systems interpretation of SAS

How can we distinguish between two algorithms with different SAS? Can the algorithm dynamics provide clues to its SAS? These questions are of practical importance because a more (statistically) stable algorithm generalizes better (via Theorem 1). Since the SAS coefficient $\beta$ is defined as a supremum over infinite pairs of training sets and inputs, a numerical estimation of it using its definition only gives a rough lower bound and obtaining this lower bound is also computationally expensive (see Figure 3 and section 5). Moreover, a numerical lower bound on $\beta$ does not give us a mechanistic understanding of statistical stability, which we seek here.

Here we develop an operator-theoretic explanation for what makes an algorithm statistically stable. We also derive a rough heuristic – albeit not a definitive predictor of generalization – that relates a given training dynamics to its SAS coefficient $\beta$.

### 4.1 Bounding the SAS coefficient

Before we present our bound, we discuss why we must deviate from the trajectory-based approach that is commonly employed to study algorithmic stability. We recall that Hardt et al. [2016] base their proofs of classical algorithmic stability (Theorems 3.7 and 3.8 in Hardt et al. [2016]) on the accumulating differences between two orbits corresponding to $S$ and $S'$, whenever the different element is chosen in the mini-batch. The greater the difference in the weights, the greater the difference in the (upper bound on the) loss functions at those weights, in the case of Lipschitz loss. Thus, Hardt et al. [2016] argue that training faster (or stopping earlier) will lead to stabler algorithms. Since SAS is about robustness of loss statistics or *infinitely* long time averages, the analysis à la Hardt et al. [2016] generically leads to vacuous bounds for the SAS coefficient.

Our analysis is based on the realization that an algorithm can be SAS even if two orbits corresponding to $S$ and $S'$ diverge from each other (within $\mathbb{W}$), but lead to similar loss statistics. Thus, to bound the SAS coefficient, a perturbation analysis of transition operators (global information), rather than a finite-time perturbation analysis of an orbit (local information), is needed. Since SAS only demands robustness on loss space, we consider Markov transition operators on the loss space ($\subseteq [0, L]$), as opposed to on the weight space ($\mathbb{W}$). In general, at any $z$ and $w_0$, the loss process, $\{\ell(z, w_t)\}_t$ is not Markovian. But, a family of Markov operators can be associated with the family of loss functions.

**Lemma 1.** *(Markov operators) Let $\mu_S$ be an ergodic, invariant measure for $\phi_S$. Assume that a loss function $\ell(z, \cdot) : \mathbb{W} \to I_z \subseteq [0, L]$ is such that the pushforward, $\nu_S^z : \mathcal{B}(I_z) \to \mathbb{R}^+$, of $\mu_S$ on the loss space is well-defined. Here, $\mathcal{B}(I_z)$ is the Borel sigma algebra on $I_z$ and $\nu_S^z = \ell(z, \cdot)_\sharp \mu_S$, where $\sharp$ refers to the pushforward operation. Then, there exists a uniformly ergodic Markov operator $\mathcal{P}_{\mu_S}^z$ on the space of probability measures on $I_z$, with its invariant measure being $\nu_S^z$.*

This lemma (see Appendix C for a proof) parallels Chekroun et al. [2014]'s Theorem A, but in a quite different setting, without using existence of a unique physical measure on the phase space ($\mathbb{W}$).

Combined with Assumption 1, perturbation results on the Markov operators, $\mathcal{P}^z_{\mu_S}$, defined by Lemma 1 lead us to SAS. Since $\mathcal{P}^z_{\mu_S}$ is uniformly ergodic, there exist $\lambda_z \in (0,1)$ and $C > 0$ such that for all $\xi \in I_z$, in the Wasserstein norm ($\|\cdot\|_W$),

$$\|\mathcal{P}^{z^t}_{\mu_S}\delta_\xi - \nu^z_S\|_W \leq C\lambda^t_z. \tag{10}$$

Let $\lambda := \sup_z \lambda_z$ be the rate of mixing associated with the family of operators $\mathcal{P}^z_{\mu_S}$. We leverage the perturbation theory of Markov operators to study the effect of stochastic perturbations on $\nu^z_S$, and subsequently, SAS. To see the connection with SAS, we recall that, by Assumption 1, for any choice of $\mu_S$, we have that $\langle \ell_z \rangle_S = \mathbb{E}_{\xi \sim \nu^z_S} \xi$. Thus, given any pair of $\mu_S$ and $\mu_{S'}$, we have (see Appendix C for details)

$$|\langle \ell_z \rangle_S - \langle \ell_z \rangle_{S'}| = |\mathbb{E}_{\xi \sim \nu^z_S} \xi - \mathbb{E}_{\xi \sim \nu^z_{S'}} \xi| \leq \|\nu^z_S - \nu^z_{S'}\|_W. \tag{11}$$

Thus, a uniform upper bound on $\|\nu^z_S - \nu^z_{S'}\|_W$ gives an upper bound for the SAS coefficient, $\beta$. Such a bound is obtained from the straightforward use of an appropriate perturbation result on uniformly ergodic Markov chains.

**Theorem 2.** *Given a uniformly ergodic Markov operator $\mathcal{P}^z_{\mu_S}$ constructed in Lemma 1, and that the Lipschitz constant of $\nabla \ell(\cdot, w)$ on $\mathbb{R}^d$ is bounded above by $L_D$ for all $w \in \mathbb{W}$, $\phi_S$ is SAS with stability coefficient*

$$\beta = \mathcal{O}\left(\frac{1}{n}\frac{L_D}{1-\lambda}\right),$$

*where $\lambda = \sup_z \lambda_z$ is the supremum over $z$ of the mixing rates of $\mathcal{P}^z_{\mu_S}$.*

*Proof.* (Sketch) The proof relies on the perturbation bounds of Rudolf and Schweizer [2018], Corollary 3.2. Note that when (10) is satisfied, and if a stochastic perturbation $S'$ introduces a change $\delta \mathcal{P}^z_{\mu_S}$ to $\mathcal{P}^z_{\mu_S}$, the perturbation bound from [Rudolf and Schweizer, 2018] gives

$$\|\nu^z_S - \nu^z_{S'}\|_W \leq C\|\delta\mathcal{P}^z_{\mu_S}\|/(1-\lambda_z), \tag{12}$$

where $\|\delta\mathcal{P}^z_{\mu_S}\|$ is the operator norm of $\delta\mathcal{P}^z_{\mu_S}$ induced by Wasserstein norm. From (11),

$$|\langle \ell_z \rangle_S - \langle \ell_z \rangle_{S'}| \leq C\|\delta\mathcal{P}^z_{\mu_S}\|/(1-\lambda_z). \tag{13}$$

Since this holds for all $z$, the right hand side of (13) is an upper bound for the stability coefficient $\beta$ in Definition (2), by taking the supremum over $z$. An upper bound on $\|\delta\mathcal{P}^z_{\mu_S}\|$ can be derived when the Lipschitz constant of $\nabla \ell(\cdot, w)$ is uniformly (in $w$) bounded on $\mathbb{R}^d$, as shown in Appendix C). $\quad\square$

Theorem 2 implies that *an algorithm that exhibits faster convergence to the stationary measure on the loss space generalizes better*. The bound in Theorem 2 implies that smaller $\lambda$ (faster convergence of the loss to the ergodic invariant measure) yields smaller upper bounds on $\beta$ (more statistically stable algorithm) and better generalization.

Strictly speaking, to determine $\lambda_z$, we must consider finite-dimensional approximations of the operator $\mathcal{P}^z_{\mu_S}$ (as done in the setting of chaotic systems in e.g., [Crimmins and Froyland, 2020, Chekroun et al., 2014]) and compute its spectral decomposition. However, in our setting, it is difficult in practice to generate samples in the loss space that obey the Markov process defined in Lemma 1. Instead, we argue for using the readily available non-Markovian loss process to estimate convergence to the equilibrium distribution. In particular, we hypothesize below that the auto-correlation function of the test loss can separate algorithms based on their SAS.

Dropping the superscripts $z$, we define the auto-correlation $C_\ell(\tau)$ in a loss function $\ell$ (see also Remark 3 in Appendix C) by

$$C_\ell(\tau) := |\lim_{T\to\infty} \frac{1}{T}\sum_{t\leq T} \ell(w_t)\ell(w_{t+\tau}) - \langle\ell\rangle^2|/\langle\ell\rangle^2. \tag{14}$$

That is, $C_\ell(\tau)$ gives the correlation between the random variables $\ell(w_t)/\langle\ell\rangle$ and $\ell(w_{t+\tau})/\langle\ell\rangle$, where the randomness comes from the batch selection $\Xi_t$ and the initialization of weights in SGD, and only from the latter in GD.

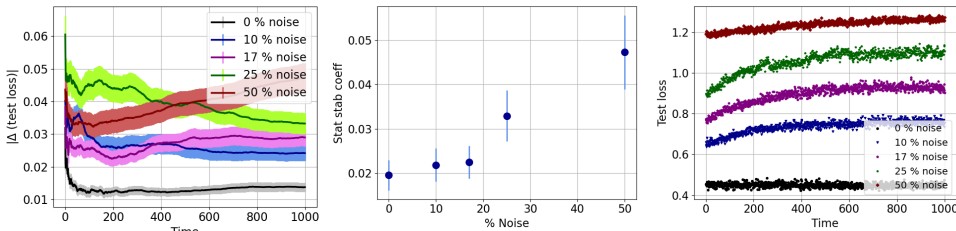

Figure 2: SAS computed from the VGG16 model. Left: The change in cumulative time averages of the test loss upon stochastic perturbation of the CIFAR10 dataset. Center: Lower bound on the statistical stability coefficient $\beta$ (Definition 2) computed using the test loss. Right: Test error timeseries.

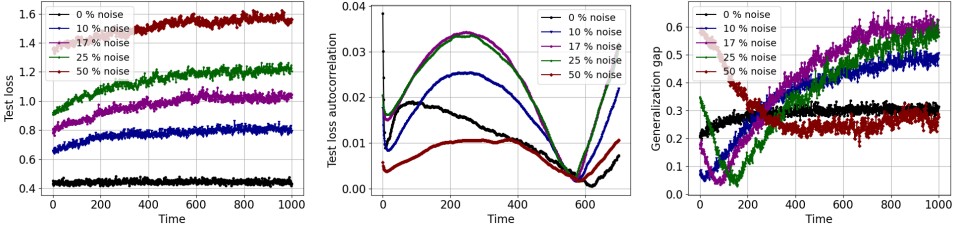

Figure 3: Predictors of generalization gaps on data with corrupted labels computed on the ResNet18 model. Left: Time series of the test loss. Center: Normalized auto-correlation of the test loss timeseries on the left. The magnitude of the auto-correlation in the test loss is suggestive of the *generalization gap* (right). For large percentage of noise, the gap decreases because the neural network cannot fit the training data well.

Suppose that the loss process is Markov, i.e., $\ell(w_{t+1})$ conditioned on $\ell(w_t)$ is independent of $\{\ell(w_{t'}) : t' \le t-1\}$. Setting $t = 0$ after a sufficiently long runup time, the function $C_\ell(\tau)$ in (14) is the auto-correlation function of a stationary Markov chain, which typically decays exponentially with $\tau$. This decay rate is lower (slower correlation decay/longer correlation times) when $\lambda$ is closer to 1. Qualitatively, comparing two different stationary Markov chains, the one with higher values of $C_\ell$ has longer correlation times and hence, larger $\lambda$.

In general, however, the loss process $\{\ell(w_t) : t \ge 0\}$ is non-Markovian. Thus, we do not expect $C_\ell(\tau)$ to decay with $\tau$. This is because the correlation function $C_\ell$ encodes both the Markovian and non-Markovian components component of the loss dynamics (see Remark 4 in Appendix C; [Zwanzig, 2001, Kondrashov et al., 2015]). Hence, we expect the correlation times in the loss process to not be equal to that of a Markov process generated according to $\mathcal{P}_{\mu_S}$. For the purpose of qualitatively comparing the SAS coefficients of two algorithms, however, the loss process can be considered a heuristic for a Markov process generated according to $\mathcal{P}_{\mu_S}$. Then, higher values of $C_\ell$ computed from the loss process indicate larger $\lambda$, and hence less statistical stability.

## 5 Numerical results

We numerically validate the main ideas of section 3 and 4 on VGG16 and ResNet18 models trained on the CIFAR10 dataset (see Appendix D for further numerical results, [Chandramoorthy and Loukas, 2023] for the code). For all our experiments, $\phi_S$ is an SGD update with momentum 0.9, fixed learning rate 0.01 and batch size of 128. In all figures, "time" indicates number of epochs. We generate different versions of the training set $S_p$ by corrupting CIFAR10's labels with probability $p$, with $S_0$ being the original CIFAR10 dataset. Figures 2 and 3 show results corresponding to $p = 0, 0.1, 0.17, 0.25$ and $0.5$. Each line in Figure 3 is a sample mean over 10 random initializations.

In Figure 2, we numerically estimate a lower bound on the SAS coefficient $\beta$ using its definition (see Definition 2). On the left, we plot the difference in test loss time-averages between between random orbits of $\phi_{S_p}$ and $\phi_{S'_p}$, with $S'_p$ being a stochastic perturbation of $S_p$. The mean over 45 pairs of orbits is shown as dots and the error bars indicate the standard error in mean. The cumulative time

average along orbits of length 1200 epochs are used as estimators for statistics. With these estimators for statistics, in Figure 2(center), we compute an estimate of the stability coefficient as the difference of the (estimated) test loss statistics at different values of $p$. This is hence an estimate on the lower bound of $\beta$, and clearly increases with $p$. This illustrates that cases with worse generalization errors (Figure 2 (right)) have larger lower bounds on $\beta$.

In Figure 3 (left), we show the test loss timeseries obtained with the ResNet18 model. Again, greater the noise corruption $p$, the larger is the generalization error (estimated by test error), consistent with previous studies Zhang et al. [2021a], Loukas et al. [2021]). On the other hand, the generalization gap – difference between training and test error – is bigger for intermediate levels of noise and decreases when $p = 0.5$, as shown in Figure 3 (center). The gap does not always increase with $p$ because, when the labels are close to random, the network cannot fit the training data and thus the training error is also large.

In Figure 3 (right), we plot $C_\ell(\tau)$ (Section 4) where $\ell$ is taken as the test loss. The test loss statistic $\langle \ell \rangle$ is estimated as a time average over 1200 epochs. At each $p$, we show the sample average of the auto-correlations over 10 independent runs. We see that the magnitude of the test loss auto-correlations preserves the same order (across $p'$s) as the generalization gap (absolute difference in test and training losses) shown in Figure 3(center). Hence, we empirically observe that the loss process codifies the phenomenological explanation for SAS that is expressed in Theorem 2.

## 6   Discussion and conclusion

Predicting generalization is an active area of research and has implications for more reliable use of machine learning. In this work, we introduce statistical algorithmic stability (SAS) and show how it implies generalization bounds. Here, we add a few important remarks.

*Statistical stability only requires robustness of statistics on loss space.* A central challenge in the analyses of non-convergent algorithms is the fact that there may be multiple, very different invariant measures $\mu_S$. For this reason, our stability criterion focuses on statistics of the loss function. This way, an algorithm can be stable even if the measures $\mu_S$ and $\mu_{S'}$ are not close in the total variation norm. In fact, an algorithm can be stable even if $\mu_S$ and $\mu_{S'}$ are mutually singular, if the loss functions have similar statistics. Since our SAS analysis hinges on a reasonable yet nonrestrictive assumption on ergodic properties of the algorithm, we believe it is broadly applicable.

*Algorithms that converge are special cases of the above analysis.* In Appendix E, we show that the proposed relationship (section 4) between statistical stability and convergence rates to stationary measures can also be observed in the Neural Tangent Kernel (NTK) regime [Jacot et al., 2018]. In this case, the training dynamics, $\phi_S$, can be approximated by a linear function of the weights, which converges to a fixed point. Thus, this provides an alternative, ergodic theoretic, interpretation of generalization in the NTK regime [Arora et al., 2019b, Montanari and Zhong, 2020, Bartlett et al., 2021].

*Broader view.* While our focus is on algorithmic stability-based generalization, this work gathers more evidence to support the broader view [Wojtowytsch, 2021, Zhang et al., 2022] that exploiting theoretically and empirically available dynamical information about the training algorithm is a fruitful complement to understanding generalization from the optimization landscape and learning theory perspectives.

**Funding:** This work was partially funded by the NSF AI Institute TILOS, NSF award 2134108, and ONR grant N00014-20-1-2023 (MURI ML-SCOPE).

**Acknolwedgments:** N.C. would like to thank Nandhini Chandramoorthy and Derek Lim for helping with the experimental setup and Sven Wang, Benjamin Zhang, Matt Li and Youssef Marzouk for valuable discussions. The authors also thank the reviewers for their constructive suggestions.

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
