# Supplementary material for "On the generalization of learning algorithms that do not converge"

**Nisha Chandramoorthy**
Institute for Data, Systems and Society
Massachusetts Institute of Technology
nishac@mit.edu

**Andreas Loukas**
Prescient Design
Genentech, Roche
andreas.loukas@roche.com

**Khashayar Gatmiry**
Electrical Engineering and Computer Science
Massachusetts Institute of Technology
gatmiry@mit.edu

**Stefanie Jegelka**
Electrical Engineering and Computer Science
Massachusetts Institute of Technology
stefje@mit.edu

## 1 Bifurcation analysis of smooth and non-convex optimization

In this section, we discuss some examples of non-convex optimization in one dimension performed with gradient descent (GD). We illustrate that, with increasing learning rate, the asymptotic behavior of orbits may alter from being periodic to quasiperiodic to chaotic. These qualitative changes are brought about through period-doubling *bifurcations*, which are observed in many physical systems (e.g., [Quail et al., 2015, Aron and Schwartz, 1984, Zhao et al., 2004]). We consider smooth and non-convex objective functions of the form $\ell_s = g_s \circ g_s \circ g_s$, with $g_s$ being the canonical quadratic map $g_s(w) := 1 - sw(1-w)$ of the unit interval $[0,1]$. Since $g_s$ is smooth, the compositions of $g_s$ with itself are smooth. The first bifurcation point appears just above $\eta = 2/\|d^2\ell_s/dw^2\|$, which is the stability threshold for convex optimization [Nesterov, 2003].

In Figure 1, the left, center and right columns correspond to $s = 1, 3$ and $4$ respectively. In the first row, we plot the loss functions $\ell_s$, which are non-convex (with multiple global minima) at $s = 3$ and $s = 4$. The second row shows the *sharpness* – absolute value of the second derivative, $a(w) = |d^2\ell_s(w)/dw|$. The third row of Figure 1 is a *bifurcation* diagram, which shows the *attractor* on the $y$-axis as a function of the learning rate. The attractor is approximated by the asymptotic orbits of the dynamics at multiple (100) different initial conditions chosen uniformly on the unit interval. Note that at small values of $\eta < 2/\|a\|$ with $\|a\| := \sup_{w \in [0,1]} a(w)$ orbits from different initial conditions converge to fixed points corresponding to the local/global minima at each value of $s$. Periodic orbits emerge at $\eta > 2/\|a\|$, that are ultimately shown to become chaotic for larger learning rates. This can be noted from the last row of Figure 1, where Lyapunov exponents (see e.g., [Katok and Hasselblatt, 1997, Wilkinson, 2017]) computed at different initial conditions are plotted. Given the gradient descent dynamics,

$$\phi_s(w) = w - \eta\ell_s'(w), \tag{1}$$

where $f'(w) = (df/dw)(w)$, the *Lyapunov exponent*, $\lambda_s : [0,1] \to \mathbb{R}$ is defined as,

$$\lambda_s(w) = \lim_{T \to \infty} \frac{1}{T} \sum_{t=0}^{T-1} \log |\phi_s'(\phi_s^t w)|.$$

Roughly speaking, this function $\lambda_s(w)$ measures the asymptotic stability of infinitesimal linear perturbations along the orbit of $w$. Here, we use the exponential notation, $\phi_s^t w = \phi_s \circ \phi_s^{t-1} w$ to denote compositions of $\phi_s$. A positive Lyapunov exponent indicates dynamical instability, e.g., chaotic orbits. We see from the bottom row of Figure 1 that in the range of learning rates considered,

36th Conference on Neural Information Processing Systems (NeurIPS 2022).

chaos is observed (positive Lyapunov exponents starting from uniformly random initial conditions) in the case of sharper minima, for larger learning rates. In ergodic systems, $\lambda_s(w)$ is independent of $w$. Indeed, for larger learning rates, we see that $\lambda_s$ appears to be independent of the initial conditions (the bottom row of Figure 1 shows $\lambda_s(w)$ for 100 different $w$). On the other hand, smaller learning rates where convergence to fixed points or periodic behavior is observed, the Lyapunov exponent converges to different negative values depending on the initial condition. This example supports the heuristic explanation for the primary assumption made in the main text about the asymptotic dynamics of learning algorithms. The assumption of lack of ergodicity ensures that the analysis is applicable to a range of learning rates, even those smaller than $2/\|a\|$.

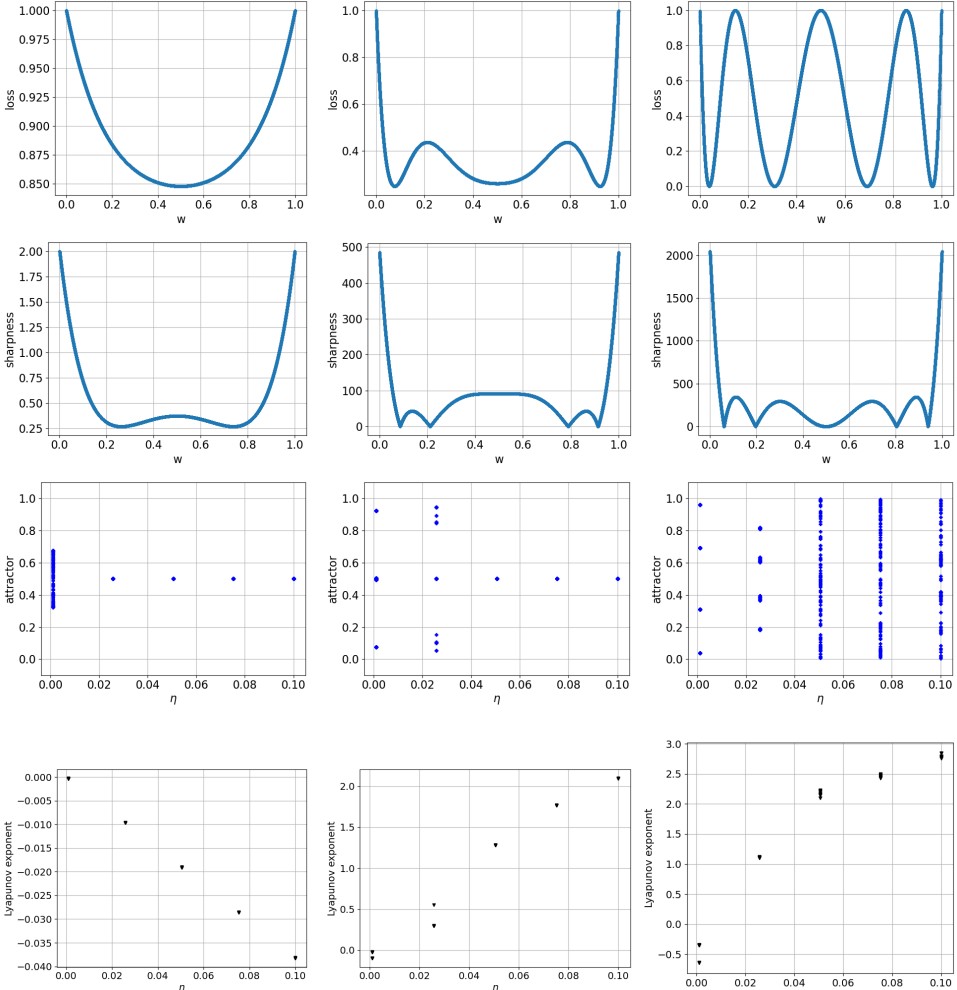

Figure 1: Bifurcation diagram of the toy model in section 1. The left column corresponds to the dynamics $\phi_s$ ((1)) at $s = 1$, the center column at $s = 3$ and the right column at $s = 4$. First row: The loss function $\ell_s$. Second row: Sharpness defined in section 1. Third row: attractors starting from multiple different initial conditions, which turn out to be fixed points at small learning rates and periodic, quasiperiodic and chaotic orbits at larger learning rates ($\gg 2/$ sharpness). Fourth row: Lyapunov exponent computed along different orbits showing chaos at large learning rates.

**Remark 1.** *Increasing the learning rate may induce ergodicity.* Assumption 1 may hold for all continuous functions for sufficiently large constant learning rates, as long as orbits do not diverge. Intuitively, large learning rates can cause more frequent transitions from the basin of attraction of one local minimum to another. On the other hand, for small learning rates, we may be able to detect the presence of multiple attractors, since different initial conditions lead to different long-time averages, as illustrated in the above toy example.

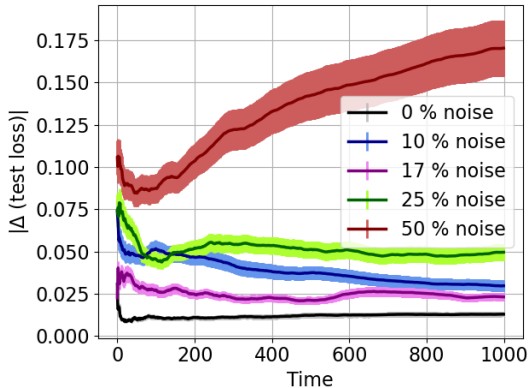

Figure 2: Absolute difference in time-averaged test loss due to stochastic perturbation computed with a ResNet18 architecture. The mean over 45 pairs of stochastic perturbations along with the standard error in mean are shown.

## 2 Proof of Theorem 1

In this section, we prove Theorem 1 from the main text. This result says that the statistical stability of an algorithm implies generalization. Recall that the population risk is defined as

$$R_S = \mathbb{E}_{z\sim\mathcal{D}}\langle\ell_z\rangle_S,$$

and the empirical risk is

$$\hat{R}_S = \frac{1}{n}\sum_{i=1}^{n}\langle\ell_{z_i}\rangle_S,$$

where the sample set $S = \{z_i : 1 \le i \le n\}$. We closely follow the proof strategy of Bousquet and Elisseeff [2002] (see also [Bousquet et al., 2020]). Define a function $\Phi_S := R_S - \hat{R}_S$, whose expected value is

$$\mathbb{E}_{S\sim\mathcal{D}^n}[\Phi_S] = \mathbb{E}_{S\sim\mathcal{D}^n}[R_S] - \mathbb{E}_{S\sim\mathcal{D}^n}[\hat{R}_S]$$

$$= \mathbb{E}_{S\sim\mathcal{D}^n}\mathbb{E}_{z\sim\mathcal{D}}\langle\ell_z\rangle_S - \frac{1}{n}\sum_{i=1}^{n}\mathbb{E}_{S\sim\mathcal{D}^n}\langle\ell_{z_i}\rangle_S \qquad (2)$$

Examining the second term, since $z_i$'s are chosen i.i.d. according to $\mathcal{D}$, $\mathbb{E}_{S\sim\mathcal{D}^n}\langle\ell_{z_i}\rangle_S$ is constant across $i$ and equal to $\mathbb{E}_{S\sim\mathcal{D}^n}\langle\ell_{z_1}\rangle_S$, where $z_1 \in S$. As in the main text, let $S'$ denote any set that has at most one element different from $S$, i.e., a stochastic perturbation of $S$. Using the stochastic perturbation $S'$, we can rewrite $\mathbb{E}_{S\sim\mathcal{D}^n}\langle\ell_{z_1}\rangle_S$ as $\mathbb{E}_{S\sim\mathcal{D}^n}\mathbb{E}_{z\in\mathcal{D}}\langle\ell_z\rangle_{S'}$. Substituting this equivalent expression in (2), and using the fact that $\phi_S$ is SAS with stability coefficient $\beta$:

$$|\mathbb{E}_{S\sim\mathcal{D}^n}[\Phi_S]| \le \mathbb{E}_{S\sim\mathcal{D}^n}\mathbb{E}_{z\sim\mathcal{D}}|\langle\ell_z\rangle_S - \langle\ell_z\rangle_{S'}|$$

$$\le \mathbb{E}_{S\sim\mathcal{D}^n}\sup_{z\in\mathbb{R}^d\times\mathbb{R}}|\langle\ell_z\rangle_S - \langle\ell_z\rangle_{S'}| \le \beta. \qquad (3)$$

The proof is based on applying Mcdiarmid's inequality (see e.g., Bousquet et al. [2020]) to $\Phi_S$, which gives a high-probability upper bound on $\Phi_S - \mathbb{E}(\Phi_S)$ in terms of the deviation of $\Phi_S$ from a $\Phi_{S'}$. In order to obtain an upper bound for the latter quantity, note that

$$|\Phi_S - \Phi_{S'}| \le |R_S - R_{S'}| + |\hat{R}_S - \hat{R}_{S'}|. \qquad (4)$$

Considering the difference of the population risks,

$$|R_S - R_{S'}| \le \mathbb{E}_{z\sim\mathcal{D}}|\langle\ell_z\rangle_S - \langle\ell_z\rangle_{S'}| \le \beta. \qquad (5)$$

Next considering the difference of empirical risks,

$$|\hat{R}_S - \hat{R}_{S'}| \le \frac{1}{n}\Big(\sum_{z_i\in S\cap S'}|\langle\ell_{z_i}\rangle_S - \langle\ell_{z_i}\rangle_{S'}| + |\langle\ell_{z_k}\rangle_S| + |\langle\ell_{z'_k}\rangle_{S'}|\Big)$$

$$\le \frac{1}{n}\left((n-1)\beta + 2L\right), \qquad (6)$$

where, recall that $L := \sup_{z \in \mathbb{R}^d \times \mathbb{R}} \sup_{w \in \mathbb{W}} |\ell(z, \cdot)|$. Putting (5) together with (6) into (4), we obtain

$$|\Phi_S - \Phi_{S'}| \leq 2\beta + \frac{1}{n}(2L - \beta).$$

**Lemma 1.** *(Mcdiarmid's inequality) Let $Z_1, \cdots, Z_n$ be random variables taking values in $\mathcal{Z}$. A function $f : \mathcal{Z}^n \to \mathbb{R}$ is said to satisfy the bounded differences property if there exists a constant $c > 0$ such that*

$$\sup_{(z_1, \cdots, z_i, z_i', z_{i+1}, \cdots, z_n)} |f(z_1, \cdots, z_n) - f(z_1, \cdots, z_{i-1} z_i', z_{i+1}, \cdots, z_n)| < c,$$

*for all single coordinate changes. For an $f$ that satisfies the bounded differences with a constant $c > 0$, given a $\delta > 0$, with probability at least $1 - \delta$,*

$$|f(Z_1, \cdots, Z_n) - \mathbb{E}[f(Z_1, \cdots, Z_n)]| \leq c\sqrt{\frac{n}{2}\log(2/\delta)}$$

Applying Mcdiarmid's inequality to $\Phi_S$, and recalling (3), we obtain the generalization bound stated in Theorem 1 of the main text. That is, with probability $\geq 1 - \delta$,

$$|\Phi_S - \mathbb{E}[\Phi_S]| = |R_S - \hat{R}_S - \mathbb{E}[\Phi_S]| \leq (2\beta + \frac{1}{n}(2L - \beta))\sqrt{\frac{n}{2}\log(2/\delta)}. \tag{7}$$

The above proof may also be repeated using the concentration inequality obtained in Theorem 4 of Bousquet et al. [2020]. This leads to a tighter bound analogous to Corollary 8 of Bousquet et al. [2020].

## 3 Predicting generalization with loss timeseries

In this section, we substantiate the connection drawn in section 4 of the main text between the rate of decay of correlations in the loss function and SAS. The study of dynamics lifted to the space of observables [Koopman, 1931] has a substantial precedent in dynamical systems theory (e.g., [Crimmins and Froyland, 2020, Keller and Liverani, 1999, Dellnitz et al., 2000]) and computational methods Arbabi and Mezic [2017], Budišić et al. [2012], Williams et al. [2014], Korda and Mezić [2018]. In particular, the idea of relating the correlation decay rate or the convergence rate of Fokker-Planck/Frobenius Perron operators with some notion of global sensitivity has also extensively appeared both in the statistical learning theory and SDE literature (see e.g. [Bartlett et al., 2021, Sirignano and Spiliopoulos, 2022]) and in the dynamical systems literature (e.g., [Kato, 2013, Chekroun et al., 2014]). In this work, inspired by the existence of such relationships in various contexts, we are able to show that the autocorrelations in the loss function can serve as predictors for the generalization gap.

This section provides the complete setting for the results in section 4 of the main text and completes the proof of Theorem 2. First we define the transition operator for probability distributions on the weight space $\mathbb{W}$ and analogous operator on the loss space.

**Markov operator for weight space.** Recall that $\Xi_t$ is a batch of $m$ indices chosen uniformly from the set $[n] := \{1, \cdots, n\}$. For GD, $\Xi_t$ is deterministic and equal to the set $[n]$. Let $(\Omega, \Sigma, \mathbb{P})$ be a probability space. Let $\mathcal{K}_S : \mathbb{W} \times \mathcal{B}(\mathbb{W}) \to \mathbb{R}^+$ be the Markov kernel associated with the update $\phi_S$, i.e., for a Borel subset $A \in \mathcal{B}(\mathbb{W})$ and a point $w \in \mathbb{W}$,

$$\mathcal{K}_S(w, A) = \mathbb{E}_{\Xi_t}\mathbb{P}(\phi_S(w) \in A | \Xi_t), \tag{8}$$

where $\mathbb{P}(\phi_S(w) \in A | \Xi_t)$ is the probability of the event that $\phi_S(w) \in A$ when the $m$ indices $\Xi_t$ are chosen. Correspondingly, we may define the Markov operator $\mathcal{P}_S$, also called the Frobenius-Perron operator Lasota and Mackey [1998], on the space of probability measures on $\mathbb{W}$,

$$\mathcal{P}_S \mu(A) = \mathbb{E}_{w \sim \mu}[\mathcal{K}_S(w, A)]. \tag{9}$$

From the above definition, it is clear that any $\phi_S$-invariant probability measure $\mu$ is an eigendistribution of $\mathcal{P}_S$ with eigenvalue 1. In our setting (see section 2 of the main text), there are potentially multiple eigendistributions corresponding to eigenvalue 1. Each invariant measure $\mu_S$ also defines different transition probabilities on the weight space $\mathbb{W}$:

$$P_{\mu_S}(A, B) = \frac{\mathbb{E}_{\mu_S}[\mathbb{1}_A \mathcal{K}_S(\cdot, B)]}{\mu_S(A)}. \tag{10}$$

**Markov operator for the loss.** Now, instead of the phase space $\mathbb{W}$, for each $z$, consider the image, $I_z \equiv \ell(z, \mathbb{W}) \subseteq [0, L]$ of $\mathbb{W}$ under $\ell(z, \cdot)$, with $L$ as defined in Theorem 1. Let $\mathcal{B}(I_z)$ denote the Borel sigma algebra on $I_z$. Analogous to the kernel (8), we may now define a kernel $\mathcal{K}_{\mu_S}^z$, for a $(\xi, E) \in I_z \times \mathcal{B}(I_z)$, as

$$\mathcal{K}_{\mu_S}^z(\xi, E) = P_{\mu_S}(\ell(z, \cdot)^{-1}(\xi), \ell(z, \cdot)^{-1}(E)). \tag{11}$$

This, in turn, gives rise to a Markov operator analogous to the Frobenius-Perron operator (9) on the full-dimensional weight space:

$$\mathcal{P}_{\mu_S}^z \nu(E) = \mathbb{E}_{\xi \sim \nu}[\mathcal{K}_{\mu_S}^z(\xi, E)]. \tag{12}$$

Note that this operator is well-defined when the level sets of $\ell_z$ are $\mu_S$-measurable. One sufficient condition for this is when the foliation of $\mathbb{W}$ by these level sets is subordinate to a measurable partition. We may then consider disintegrations of $\mu_S$ on this measurable partition and define the kernel (11) using conditional measures supported on elements of the partition. Further, it is clear that $\mathcal{P}_{\mu_S}^z$ satisfies the properties of a Markov operator (positive unity preserving contraction).

Thus, in order to prove Lemma 1, it remains to show that the operator defined by (12) is mixing. For this, we make an additional assumption. We assume that the Frobenius-Perron operator $\mathcal{P}_S$ mixes to the measure $\mu_S$ starting from any measure of the form $\mu = \ell(z, \cdot)^{-1}\nu$, for an absolutely continuous probability measure $\nu$ on $[0, L]$. That is,

$$\|\mathcal{P}_S^t \mu_S - \mathcal{P}_S^t \mu\|_{\text{TV}} = \mathcal{O}(\zeta^t \|\mu_S - \mu\|_{\text{TV}}), \tag{13}$$

where $\| \cdot - * \|_{\text{TV}}$ indicates the total variation distance and $\zeta \in (0, 1)$ is the rate of mixing. On the other hand, since $\mathcal{P}_S^t \mu \to \mu_S$ weakly, $\nu_t^z := \left(\mathcal{P}_{\mu_S}^z\right)^t \ell(z, \cdot)_\sharp \mu \to \nu_S$ weakly on $\mathcal{B}(I_z)$. Intuitively, we expect that rate of mixing of the latter, say $\lambda^z \in (0, 1)$ correlates with $\zeta$ since

$$\left|\langle \ell_z \rangle_S - \mathbb{E}_{\xi \sim \ell(z, \cdot)_\sharp \mu_t}[\xi]\right| \leq \sup_{f, \|f\| \leq 1} \|\mathbb{E}_{w \sim \mu_S}[f(w)] - \mathbb{E}_{w \in \mu_t}[f(w)]\|$$

$$= \|\mu_S - \mu_t\|_{\text{TV}}. \tag{14}$$

Assuming that $\mathbb{W}$ is a Polish space, the above relationship (14) conveys that when a measure $\mu_t$ converges to $\mu_S$ in the TV norm, expectations with respect to $\mu_t$ of all continuous functions, of which $\{\ell(z, \cdot)\}_z$ is a subset, also converge to expectations with respect to $\mu_S$. Thus, intuitively we expect that $\sup_z \lambda^z$ is a lower bound for $\zeta$ ((13)). In the main text, below Lemma 1, we state the uniform ergodicity of $\mathcal{P}_{\mu_S}^z$ in terms of the Wasserstein norm. This holds from (14) since convergence in TV distance implies convergence in Wasserstein. This concludes the proof of Lemma 1. Finally, note that our setting is different from previous works ([Chekroun et al., 2014] and references therein) in the dynamics literature that use observable-specific Markov operators, in that we do not assume uniqueness of the ergodic, invariant measure on the full-dimensional (weight) space.

**Effect of stochastic perturbations** In section 4 of the main text, we use the perturbation theory of mixing Markov operators to relate the SAS coefficient to the mixing rate of $\mathcal{P}_{\mu_S}^z$. This perturbation bound (from [Rudolf and Schweizer, 2018]) is given in terms of the perturbation $\delta \mathcal{P}_{\mu_S}^z$ to the operator $\mathcal{P}_{\mu_S}^z$ when a stochastic perturbation $S \to S'$ is applied to the training set. Here we discuss the size of $\delta \mathcal{P}_{\mu_S}^z$, completing the proof of Theorem 2 in the main text. First note that $\|\delta P_{\mu_S}^z\| \leq \|\mathcal{P}_S - \mathcal{P}_{S'}\|$, and hence it suffices to obtain an upper bound for $\|\mathcal{P}_S - \mathcal{P}_{S'}\|$. The perturbation to Markov kernel in the weight space due to a stochastic perturbation is given by

$$\mathcal{K}_S(w, A) - \mathcal{K}_{S'}(w, A) = \sum_{\Xi \in \Delta} \Big( \mathbb{P}(\phi_S(w) \in A|\Xi) - \mathbb{P}(\phi_{S'}(w) \in A|\Xi) \Big), \tag{15}$$

where $\Delta$ is the set of $m$ indices from $[n]$ that contain $k$, the index at which $S$ and $S'$ differ. When $\Xi$ is a uniform random variable as we have assumed, the cardinality $\Delta$ is $\binom{n-1}{m-1}/\binom{n}{m} = m/n$. This leads to the following upper bound on the perturbation size in the Wasserstein norm,

$$\|\mathcal{P}_S \mu - \mathcal{P}_{S'} \mu\|_W := \sup_{f, \|f\|_{\text{Lip}} \leq 1} |\mathbb{E}_{w \sim \mathcal{P}_S \mu} f - \mathbb{E}_{w \sim \mathcal{P}_{S'} \mu} f|$$

$$= \sup_{f, \|f\|_{\text{Lip}} \leq 1} |\mathbb{E}_\Xi \mathbb{E}_{w \sim \mu} f \circ \phi_S - \mathbb{E}_\Xi \mathbb{E}_{w \sim \mu} f \circ \phi_{S'}| \leq \frac{m}{n} \sup_{w \in \mathbb{W}} \|\phi_S(w) - \phi_{S'}(w)\|$$

$$\leq \eta \frac{m}{n} \sup_{w \in \mathbb{W}} \|\nabla L_S(w) - \nabla L_{S'}(w)\| \leq \frac{m\eta}{n^2} \sup_w \|\nabla \ell(z_k, w) - \nabla \ell(z_{k'}, w)\|$$

$$\leq c\, m\, L_D \eta / n^2. \tag{16}$$

In order to apply this bound, we use a result from the perturbation theory of Markov chains from [Rudolf and Schweizer, 2018] (Corollary 3.2 in the asymptotic limit) as explained in the proof sketch of Theorem 2 in section 4. Thus, we obtain an upper bound for $\beta = \mathcal{O}(L_D/(1-\lambda)n)$ for GD (with $m = n$) as claimed in Theorem 2.

**Remark 2.** *In this section as well as section **??**, we use inequalities between different norms on the space of finite signed measures. This is a Banach space with the total variation norm, isomorphic to $L^1(\Lambda)$ for some background measure $\Lambda$. The inequalities (for instance, (**??**)) follow from the dual characterization of the norms.*

**Remark 3.** *Although the definition of the autocorrelation in section 4 appears to be a function of $w_0$, $C_\ell(\tau)$ does not depend on $w_0$ due to Assumption **??**. Note that the rate of convergence to equilibrium, the coefficient $\lambda$ in the definition of uniform ergodicity, also determines the correlation decay rate when $\ell$ is initialized out of equilibrium (That is, $\ell(w_0)$ does not sample $\nu_S$).*

**Remark 4.** One way to understand the non-Markovian loss process is through the Mori-Zwanzig formalism [Zwanzig, 2001] that originated in statistical mechanics and has found extensive applications in deriving reduced-order models for complex physics (e.g. see [Lin and Lu, 2021, Kondrashov et al., 2015]). In this formalism, we can consider the exact evolution of a finite set of observables, $\Psi(w) = [\psi_1(w), \cdots, \psi_p(w)]$ such that $\ell \in \text{span}\{\Psi\}$ (here, $\ell := \ell_z$ for an arbitrary $z$). At time $t$, $\Psi_t := \Psi(w_t)$ can be written as a sum of three terms: 1) a Markov term that depends on the values $\Psi_{t-1}$, 2) a non-Markovian memory term that is a function of $\{\Psi_{t-k} : t \geq k > 1\}$, and 3) a noise term that is a function of $w_0$. The first two terms depend on the dynamics $\phi_S$ and are hence different for different values of $S$. The autocorrelation function $C_\ell$ has contributions from both the Markovian and non-Markovian terms.

# 4 Stability experiments on ResNets

We obtain similar results for SAS with the ResNet18 model as with the VGG16 model shown in Figure 3 of the main text (see [Chandramoorthy and Loukas, 2023] for the code). In Figure 2, we plot the difference in the cumulative average of the test loss at runs with the ResNet18 architecture. The difference is taken between two SGD runs with the same parameters as in section 5 of the main text and with training data that are stochastic perturbations of each other. We consider 45 pairs of stochastically perturbed datasets (see section 2 of main text for definition of stochastic perturbation) each for each value of $p$. The value of $p = 0, 0.1, 0.17, 0.25, 50$ indicates the probability of error injected into the labels of the CIFAR10 dataset. The mean of the absolute difference in the cumulative test loss is shown in dark colors while the standard error in mean in the corresponding lighter color. The time averages are calculated over 1200 epochs after a run up time of 200 epochs. The results indicate that greater the noise probability $p$, greater the estimate of SAS (less statistical stability). Hence, these results indicate that statistical algorithmic stability (see section 2) correlates with the generalization performance of the models at the difference label corruption (noise) levels. Furthermore, considering together with the VGG16 results presented in the main paper, this relationship between SAS and generalization holds across the different architectures we have employed in this paper.

The generalization gap plotted in Figure 3 (center) (of the main text) is the cumulative time average (ergodic average) of the absolute difference between the test and training errors. This is approximately equal to (strictly, an upper bound for) $|R_S - \hat{R}_S|$. A loose upper bound for this quantity comes from the theoretical generalization bound (in Theorem 1). In practice, this quantity is estimated to be small relative to the test error with corrupt datasets. When the noise probability is 0% (original dataset), the gap estimate is about 80% of the test error because the training error is small. But, when the noise probability is 50%, the gap estimate is about 20% of the test error since the training error is also large. We may not observe this reduction in the generalization gap if the errors (test and training) were defined using pointwise values. That is, since, $\sup_w |E_{z\sim D}\ell_z(w) - (1/n)\sum_{i=1}^n \ell_{z_i}(w)| \geq |R_S - \hat{R}_S|$, upon early stopping when training error is low, we may not observe this phenomenon.

Finally, we remark that our empirically estimated autocorrelation function serves as a proxy until a more sophisticated method for the estimation of $\lambda$ is developed. As mentioned in the main text (section 4), this is a challenging problem that is beyond the scope of this work.

## 5 Revisiting stability in the linear regime

Since its introduction by Jacot et al. [2018], training in the Neural Tangent Kernel (NTK) regime has been analyzed thoroughly in numerous works, wherein its convergence to kernel ridge regression has been formally proved in two-layer, infinitely wide networks [Montanari and Zhong, 2020, Bartlett et al., 2021], infinitely wide fully connected networks [Arora et al., 2019a], convolutional networks Arora et al. [2019b] and so on. Using its equivalence with kernel ridge regression, the generalization properties under the NT regime have also been well-studied (see Bartlett et al. [2021] for a review). Here, we revisit the NT regime with the purpose of demonstrating that the analysis in the present paper also holds when $\phi_S$ achieves a fixed point.

The Neural Tangent Kernel (NTK) model is an approximation of a neural network whose parameters remain close to initialization during training. Given that training in the NT regime is well-approximated by the dynamics of linear regression (see Theorem 3.1 and 3.2 of Arora et al. [2019b]), in order to apply our dynamics-based generalization analyses to the NTK regime, we need only consider linear regression dynamics. That is, let $\tilde{w}_t$ be an orbit of a perturbed dynamics (the linear regression dynamics, see Lemma 1 of [Arora et al., 2019b]) close to an orbit $w_t = \phi_S(w_{t-1})$ for all time, so that, for some small $\epsilon > 0$,

$$\lim_{t \to \infty} \|w_t - \tilde{w}_t\| = \|w^* - \tilde{w}^*\| \le \epsilon. \tag{17}$$

As mentioned in the main text (section 3), SAS reduces to the standard notion of algorithmic stability (see e.g., [Mohri et al., 2018] chapter 14 for a survey) when the dynamics converges to a fixed point. Now suppose that the fixed point $\tilde{w}^*$ is algorithmically stable. That is, for all stochastic perturbations $S'$ of $S$, and for all $z$,

$$|\ell(\tilde{w}_S^*, z) - \ell(\tilde{w}_{S'}^*, z)| \le \beta.$$

Assuming Lipschitz loss with Lipschitz constant $C_{\text{Lip}}^z$ and denoting by $C_{\text{Lip}} = \sup_z C_{\text{Lip}}^z$

$$\begin{aligned} |\ell(w_S^*, z) - \ell(w_{S'}^*, z)| &\le |\ell(\tilde{w}_S^*, z) - \ell(\tilde{w}_{S'}^*, z)| + |\ell(w_S^*, z) - \ell(\tilde{w}_S^*, z)| \\ &\quad + |\ell(w_{S'}^*, z) - \ell(\tilde{w}_{S'}^*, z)| \tag{18} \\ &\le \beta + 2C_{\text{Lip}}\epsilon. \tag{19} \end{aligned}$$

That is, the NTK orbit is stable with the stability coefficient $\beta + 2C_{\text{Lip}}\epsilon$. Thus, in order to prove the algorithmic stability of the NTK orbit, it is enough to consider the stability of the linear regression orbit, which is a linear dynamical system as we describe below.

The second main idea that we develop is the prediction of the stability coefficient by the rate of decay of correlations. In this case, we show that the speed of convergence to the fixed point determines the generalization properties via algorithmic stability. We derive stability-based generalization bounds, an alternative to Rademacher complexity-based bounds in Arora et al. [2019a]. Comparing with existing generalization results in the well-understood NTK regime is an ideal test bed for the alternative dynamical perspective of this present work.

**Linear dynamics** Let $w_r \in \mathbb{W}$ be the parameter at which a NN $h_{\text{NN}}(\cdot, w_r)$ is the zero function from $\mathbb{R}^d$ to $\mathbb{R}$, i.e., $h_{\text{NN}}(x, w_r) = 0$, for all $x$. Now consider training the weights $w \in \mathbb{W}$ of the NN, with the initialization $w_0 = w_r$. With GD on the squared loss $L_S(w) = (1/2) \sum_{i=1}^n (y_i - h_{\text{NN}}(x_i, w))^2$, and learning rate $\eta > 0$, the dynamics of the weights are as follows,

$$\phi_S(w) = w + \eta \, \nabla h_{\text{NN}}(X, w)^T (Y - h_{\text{NN}}(X, w)). \tag{20}$$

Here, $X = [x_1, \cdots, x_n]^T \in \mathbb{R}^{d \times n}$ and the notation $h_{\text{NN}}(X, w) \in \mathbb{R}^n$ represents $[h_{\text{NN}}(x_1, w), \cdots, h_{\text{NN}}(x_n, w)]^T$. Note that the above dynamics $\phi_S(w)$ is a nonlinear function of $w$. Now we consider the NTK setting described in Bartlett et al. [2021] so that we replace $h_{\text{NN}}$ with its linearization about $w_r$, $\nabla h_{\text{NN}}(x, w_r)(w - w_r)$. With this linearization about $w_r$, the above dynamics $\phi_S$ becomes linear in $w$,

$$\tilde{\phi}_S(w) = w + \eta \Phi_S^T (Y_S - \Phi_S(w - w_r)), \tag{21}$$

where $Y_S = [y_1, \cdots, y_n]^T \in \mathbb{R}^n$. Recalling that $\mathbb{W} \subset \mathbb{R}^{d_w}$, $\Phi_S$ is an $n \times d_w$ matrix with the $i$th row being $\nabla h_{\text{NN}}(x_i, w_r)$. In the NTK regime, the dynamics $\phi_S$ is well-approximated by $\tilde{\phi}_S$ (see Theorem 5.1 of [Bartlett et al., 2021] for conditions under which the approximation holds). That is,

the linear dynamics close to the NTK dynamics, referred to as the linear regression dynamics above, is the following,

$$\tilde{w}_{t+1} = A_S \tilde{w}_t + b_S, \tag{22}$$

where $K_S := \Phi_S^T \Phi_S$, $A_S := (I - \eta K_S) \in \mathbb{R}^{d_w \times d_w}$ and $b_S := \eta \left( K_S w_r + \Phi_S^T Y_S \right) \in \mathbb{R}^{d_w}$. The dynamics converges to a fixed point $\tilde{w}_S^* = A_S \tilde{w}_S^* + b_S$, as long as $\|A_S\| < 1$.

**Evolution on function space** Note that the invariant distribution of the above dynamics is singular: the delta distribution centered at $\tilde{w}_S^*$. In order to repeat the analysis in section 4 for this special case, we need to obtain the relationship between the rate of decay of correlations with respect to this invariant measure and the stability of the fixed point. First isolating the rate of decay of correlations, this rate is equal to the second largest eigenvalue of the associated Frobenius-Perron on $L^2(\mathbb{W})$ or equivalently, the Koopman operator on $L^2(\mathbb{W})$ (since these two linear operators are adjoint to each other, they share isolated spectra). Since the dynamical system is linear, it is easy to verify that the eigenvalues of $A_S$ are also Koopman eigenvalues. We can also check that the eigenfunctions corresponding to eigenvalue $\theta_i$ are of the form $v_i^T w + 1/(\theta_i - 1) v_i^T b_S$, where $v_i$ are left eigenvectors of $A_S$.

Note that $A_S$ is a symmetric matrix whose largest absolute eigenvalue is equal to $1 - \eta \theta_{\min}$, where $\theta_{\min} > 0$ is the smallest eigenvalue of the NTK $\hat{K}_S := \Phi_S \Phi_S^T$.

**Stability of the fixed point** In order to relate the rate of convergence, $\lambda := 1 - \eta \theta_{\min}$ with SAS in this case, we now describe SAS in this regime. As we noted previously, SAS reduces to the algorithmic stability of the fixed point for the dynamics (22). In order to deduce the stability of the fixed point to stochastic perturbations in the input, we need to recognize that the fixed point is an exact interpolant.

We can check that $\tilde{\phi}_S(\tilde{w}_S^*) = \tilde{w}_S^*$ iff $Y_S = \tilde{\phi}_S(\tilde{w}_S^* - w_r)$. That is, the function $\nabla h_{\mathrm{NN}}(x, w_r)(\tilde{w}_S^* - w_r)$ exactly interpolates at the data points, among the class of linearized functions. In other words, the fixed point is $\tilde{w}_S^* = w_r + a_S$, where $a_S$ is the minimum norm interpolation solution given by $a_S = \Phi_S^T(\Phi_S \Phi_S^T)^{-1} Y_S$. Thus, having a closed form expression for $\tilde{w}_S^*$, we can obtain an upper bound on the stability of the algorithm $\phi_S$.

Since $\hat{K}_S = \Phi_S \Phi_S^T$ and its inverse are symmetric, $\|(\hat{K}_S)^{-1}\|$ is also the maximum eigenvalue of $(\hat{K}_S)^{-1}$. A stochastic perturbation $S'$ of $S$ introduces a rank-one change denoted $\delta K$ to $(\hat{K}_S)^{-1}$. From Weyl's inequality, $\|(\hat{K}_S)^{-1} - (\hat{K}_{S'})^{-1}\| \leq \|\delta K\|$. In the case of Lipschitz loss, an upper bound on $\beta$ therefore depends on the maximum eigenvalue of $\hat{K}_S^{-1}$, which is equal to $1/\theta_{\min}$. Thus, we see that a smaller $\theta_{\min}$ implies a smaller rate of convergence (slower convergence) as well as a larger upper bound on $\beta$ (lesser algorithmic stability). Hence, the linear regime also supports the analysis in section 4, which discusses a more general scenario of convergence of weights in distribution.

**Remark 5.** *As an aside that applies to the entire paper, we clarify that by "statistics" we refer to statistics over the parameter space $\mathbb{W}$. The distribution over the weight space is specified in each context. Recall that the randomness in the sense of randomness over the weights arises due to the stochastic nature of SGD as well as the randomness over initial conditions.*

**Remark 6.** *While we have considered linearization about a point, we may repeat the above analysis by considering another linear network via modifying the definition of the empirical matrix $\Phi_S \Phi_S^T$ to $K_S = E_{w \in \mathcal{D}_w} \nabla h_{\mathrm{NN}}(X, w) \nabla h_{\mathrm{NN}}^T(X, w)$. This is the well-studied limit of the empirical kernel as the number of neurons tends to infinity.*