# OpenReview forum: "On the generalization of learning algorithms that do not converge"
_NeurIPS.cc/2022/Conference — NeurIPS 2022 Accept_

### Official Review · Reviewer_HJ1k · 2022-06-23

**Rating:** 8
**Confidence:** 3
**Soundness:** 3 good
**Presentation:** 4 excellent
**Contribution:** 4 excellent

**Summary:**

This paper, entitled "On the generalization of learning algorithms that do not converge", extends the notion of algorithmic stability to algorithms that do not converge in the traditional sense of reaching a fixed point in weight space. Because it is increasingly becoming clear that many commonly used algorithms often do not converge, this covers a broad range of algorithms. By using ergodic theory, the authors define statistical algorithmic stability and develop bounds for their metric. These bounds identify that algorithms "that exhibit faster convergence to the stationary measure on the loss space generalize better." Finally, the authors validate their theory on VGG-16 and ResNet-18 models.

**Questions:**

I have no additional questions for the authors.

**Limitations:**

The authors should make a little more clear what possible limitations the use of the auto-correlation and the non-Markovian nature of samples from the loss space introduces into their numerical experiments.

**Strengths And Weaknesses:**

STRENGTHS:

1. This paper is very well written. It was easy to read and get the main points from.

2. This paper effectively motivates the problem by identifying a gap in the current theory developed for studying algorithmic stability and illustrating why many commonly used algorithms fall into this gap. TA large body of literature was cited and it was clear where overlap between this paper's contributions and other work in the field was.

3. The authors' new definition of statistical algorithmic stability is general, intuitive, and theoretically backed. This last point, being grounded in theory, makes it especially appealing as an extension for algorithmic stability.  I believe that it will be prove useful for studying algorithmic stability, and will open up new discussion in the field.  Additionally, the authors should be applauded for their treatment of dynamical systems and ergodic theory, which was clear and well laid out.

4. The result that "that exhibit faster convergence to the stationary measure on the loss space generalize better" is very interesting. If anything, I think the authors should emphasize this point more throughout the paper, as I imagine it will be of interest to the community.

WEAKNESSES:

1. The discussion surrounding the use of the auto-correlation and the non-Markovian nature of samples from the loss space felt cramped. I understand page that there are page constraints, but it felt a little difficult to connect the experimental results with the theory that had been laid out earlier. Additionally, I would emphasize the last (punchline) sentence of Sec. 4 (lines 324-326) more.

2. The generalization gap for the 50% noise being nearly the same (if not lower) than the generalization gap of the 0% noise confuses me (Figure 2). Is this because the ResNet is performing "at chance"? Adding a sentence or two to clarify this would be helpful for interpreting this result.

3. There are two points in Sec. 5 where I believe the conclusions from the results are overstated. First, Figure 3 (right) shows the lower bounds of $\beta$. Therefore, the claim "illustrating that cases with worse generalization error have larger $\beta$" (lines 344-345), is not quite right because it has not been shown how tight the lower bound is. However, I do think it is interesting and telling that the lower bound increases with worse generalization. Making this point more accurately does not, in my mind, diminish the finding. And second, "the magnitude of the test loss autocorrelation correlates with with the generalization gap" (lines 353-354). Looking at Figure 2 (center), it appears that the generalization gap between 10%, 17%, and 25% noise, at the end of the 1000 epochs, is pretty minimal. However, there is a very clear difference in the magnitude of the test loss autocorrelation between 10% and 17%/25% (Figure 2 (right)). Similarly, there is a large difference in magnitude between 0% and 50%, whereas there is not a major difference between the generalization gap of those two. I think the authors would do better to report the results as is. The theory they develop is very convincing (at least to me), and improvements on the bounds and ways to compute $\lambda$ will lead to stronger numerical results. For now, it is sufficient (again, to me) to just show the current state of numerical results, even if they aren't 100% convincing.

3. The abstract mentions the work "provides clues to overfitting" (line 12), but overfitting is not explicitly mentioned. I would either discuss this in Sec. 6 or remove from the abstract.

MINOR POINTS:

1. Figure 3 (left) should have lines instead of dots to reduce the thickness of the curves.

2. That Figure 2 and Figure 3 are tested on different models should be made more clear in the text (not just the figure captions).

3. It should be noted after Eq. 7 that a smaller value of $\beta$ implies better generalization. This is of course obvious from the equation, but "stability coefficient" may make readers think bigger is more stable.

4. "two notions of risk defined..." (lines 250-252) is an interesting point and should be made a little more clear. What precisely is "the space of learning algorithms"? Why does Eq. 8 lie in the space?

5. There are a few typos:

i. Lines 94 and 99 have mis-formatted citations.

ii. Line 355 should say Eq. 12 or (12).

iii. Should it say "Figure 3 (left)" on line 345?

6. There are a few places where things should be defined before they are used:

i. $m$ being the batch size in line 133.

ii. $|| \cdot ||_{TV}$ in Eq. 10.

iii. The # symbol in line 274.

7. The literature was in general very well cited, but I think adding some references when discussing dynamical systems theory and ergodic theory in ML would be helpful. Eg: Saxe et al. 2013; Li et al. 2018; Dietrich et al. 2020; Dogra and Redman, 2020.

OVERALL:

I believe this paper is strong and fills an important need in the field. Given that generalization is a major problem in machine learning, and that current approaches based in algorithmic stability have limitations (namely, they require that the algorithmic converge in weight space, which many commonly used algorithms do not do). Additionally, I believe that the ergodic theory, and dynamical systems theoretic more generally, perspective the authors take is a very fruitful direction for the general field of ML to move in.

For all these reasons, I am giving the paper a 7. That being said, I do believe the points I discuss above in the weakness section (which I believe can be addressed in a straightforward manner) need to be addressed for this score to remain this high.

---

> ### Author Response · Authors · 2022-08-02
> **Response to reviewer HJ1k**
>
> We thank the reviewer for their highly thoughtful comments and thorough reading and understanding of our paper.
>
> > The discussion surrounding the use of the auto-correlation and the non-Markovian nature of samples from the loss space felt cramped. I understand page that there are page constraints, but it felt a little difficult to connect the experimental results with the theory that had been laid out earlier. Additionally, I would emphasize the last (punchline) sentence of Sec. 4 (lines 324-326) more.
>
> We completely agree. We are revising the endings of section 4 and 5 for clarity.
>
> > Is this because the ResNet is performing "at chance"? Adding a sentence or two to clarify this would be helpful for interpreting this result.
>
> Indeed, we empirically observed that both the test and training errors are large in the 50% noise case, leading to a smaller generalization gap. We added this clarifying sentence to the text in section 5.
>
> > However, I do think it is interesting and telling that the lower bound increases with worse generalization.
>
> We completely agree with your assessment. We have now revised the statement to say that the figure (now Figure 3) only illustrates that the estimator for the lower bound increases with increasing generalization error.
>
> > Similarly, there is a large difference in magnitude between 0% and 50%, whereas there is not a major difference between the generalization gap of those two.
>
> This is again an excellent observation – thank you. We do agree that we do not have a theoretical explanation for the smaller generalization gap in the case of 50% noisy labels. Our observation (mentioned above) is that the ergodic average of the training error converges to a higher value in the 50% noise case and therefore the difference between the ergodic averages of the training and test errors is estimated to be lower than with less corrupt datasets. We think that we may not observe this reduction in the generalization gap if the errors (test and training) were defined using pointwise values, as in the classical algorithmic stability literature. That is, since, $\sup_w |E_{z \sim D} \ell_z(w) - (1/n)\sum_{i=1}^n \ell_{z_i}(w)|\geq |R_S - \hat{R}_S|,$ upon early stopping when training error is low, we may not observe this phenomenon. We have added this comment to the supplementary section (Appendix D).
>
>  > ... and improvements on the bounds and ways to compute $\lambda$ will lead to stronger numerical results. For now, it is sufficient (again, to me) to just show the current state of numerical results, even if they aren't 100% convincing.
>
> This is a highly pertinent point, thank you very much. Section 4 is intended to reveal a phenomenological connection between training dynamics and stability of its statistics. As you correctly point out, we do not develop a sophisticated method to obtain the theoretically understood rate of convergence $\lambda$. This computation falls under data-driven methods development for the spectral data of the transfer/Koopman operators, which is an active area of research. Here, compared to many physical nonlinear systems, the lack of a unique ergodic, invariant measure on the phase space (space of weights) implies that data-driven methods from the dynamics literature may not be valid without modification. Thus, developing bespoke methods for $\lambda$ in this setting will be an interesting challenge to explore in a future work. We have remarked upon this in Appendix D.
>
> ### Minor points
>
> > Figure 3 (left) should have lines instead of dots to reduce the thickness of the curves.
>
> We will fix this in the final revision, thank you.
>
> >   That Figure 2 and Figure 3 are tested on different models should be made more clear in the text (not just the figure captions).
>
> We have added a clarification in section 5 of the revised version.
>
> > ... of course obvious from the equation, but "stability coefficient" may make readers think bigger is more stable.
>
> Done in the revision.
>
> >  What precisely is "the space of learning algorithms"? Why does Eq. 8 lie in the space?
>
> The space of learning algorithms can be parameterized by all the algorithmic parameters (such as the learning rate), architecture, dataset etc. What we mean is that the training and generalization errors -- $R_S$ and $\hat{R}_S$ -- defined in Eq. 8 -- are scalar functions of this algorithmic parameter space. For instance, keeping all other parameters fixed, if we vary the dataset, the training and generalization errors change. This variation is what we study in SAS. Due to the averaging over hypothesis space or weight space inherent in the definitions of $R_S$ and $\hat{R}_S$, they are not functions of the hypothesis/weight spaces, unlike the classical definitions of training and test errors.
>
> >  Other minor comments
>
> Thank you for the valuable references -- they will be added to the final revision. All other fixes have been made in the revision.

---

> > ### Comment · Reviewer_HJ1k · 2022-08-05
> > **Reply to authors**
> >
> > Thank you for your detailed and helpful comments. It was clear that you took time to consider what I asked/suggested and your response clarified many of the questions I had. As I said in the original review, I think this is very strong work and with your revisions I believe this is even more the case.
> >
> > Finally, I especially find your comment about using transfer/Koopman operators (with modification) a very exciting and pertinent idea. The fact that there are ways in which the $\lambda$ can be more accurately computed in the future only strengthens my belief that this work will be able to provide new insight into the field in the future.

---

> > > ### Author Response · Authors · 2022-08-08
> > > **Thank you!**
> > >
> > > Thank you very much for spending time to understand our paper and our rebuttal!
> > > We appreciate your time and effort.

---

### Official Review · Reviewer_mM1Z · 2022-07-10

**Rating:** 6
**Confidence:** 4
**Soundness:** 3 good
**Presentation:** 4 excellent
**Contribution:** 3 good

**Summary:**

This paper lays the theoretical groundwork for studying the behavior (generalization, dynamics) of stochastic learning algorithms: here, the appropriate notion is not deterministic convergence to a fixed-point, but rather various notions of stochastic convergence. Specifically, the paper posits (Assumption 1) that the dynamics of the hypothesis (input-output function) are ergodic. Thus, functionals of the hypothesis may converge over the SGD trajectory, while the parameters themselves may not converge. The paper also discusses relations to existing notions of stability and convergence, and provides a toy analysis for when SGD learning is "stochastically stable."


**Questions:**

See above.

**Strengths And Weaknesses:**

Strengths:
The paper is very well-motivated and well-written: it correctly points out limitations of prior (deterministic) approaches to analyzing learning dynamics, and it clearly outlines a formalism for studying stochastic dynamics of SGD. The mathematical exposition is clear.

Weaknesses:
Assumption 1 is very strong, and is not tested with proper experiments. For example: Assumption 1 (and the definition of SAS) involves a supremum over all inputs z, and all test functions \ell. However, the experiments in Figure 1 only measure several observables, and make no effort to take the supremum over z.
This limitation should be discussed in the paper: do the authors claim that Assumption 1 is realistic, or is it merely meant to be theoretically-suggestive, while being too strong to hold in practice?

I am also curious whether Assumption 1 is consistent with the "Disagreement Property" in the literature (introduced by [1][https://arxiv.org/abs/2009.08092] and followed-up by [2][https://arxiv.org/abs/2106.13799]).
The Disagreement Property in [2] shows that two models trained with different seeds, on the same train dataset, *do not* converge to the same hypothesis function. Rather, the two hypotheses "disagree" in their predictions, at a rate roughly equal to the test error.
This property seems at-odds with the convergence that's claimed in the present paper, and I'd be curious to hear the author's thoughts on this.

---

> ### Author Response · Authors · 2022-08-02
> **Response to Reviewer mM1Z**
>
> We thank the reviewer for a careful reading of our paper and their constructive questions on our theoretical setting.
>
> > Assumption 1 is very strong, and is not tested with proper experiments. For example: Assumption 1 (and the definition of SAS) involves a supremum over all inputs z, and all test functions \ell. However, the experiments in Figure 1 only measure several observables, and make no effort to take the supremum over z.
> > Do the authors claim that Assumption 1 is realistic, or is it merely meant to be theoretically-suggestive, while being too strong to hold in practice?
>
> The essence of Assumption 1 is that the initialization of weights does not affect the *distribution* of loss values over a long orbit, even though it affects the asymptotic probability distribution in the space of weights. We have empirically tested this over tens of different initializations on two different models (VGG16 and ResNet18) by taking time averages of test losses and observing that they are almost constant. While, of course, this does not rule out that the assumption may be violated in some settings, we have not seen evidence for the violation of this assumption in practice in our experiments.
>
> Assumption 1 is challenging to verify numerically because it must hold for an infinite family of loss functions (at each input $z$). It is computationally expensive to verify ergodicity even over a finite class of loss functions (at a set of data points, as we have done in figure 1) because we must obtain ergodic averages precisely. These are random variables when the length of the orbit is finite and thus there is noise in their estimation.  Roughly, from the central limit theorem, the variance in the estimate decays as ${\cal O}(1/T)$, where $T$ is the number of epochs used to compute the average.
>
> We would like to end by stressing that, even if Assumption 1 may not always hold in practice, it constitutes a significant improvement to currently leading theories: we relax the assumption of point-wise convergence in the weight space to that of convergence in distribution of the function space which constitutes an improvement in two aspects (distribution vs point-wise and hypothesis vs weights).
>
> > The Disagreement Property in [2] shows that two models trained with different seeds, on the same train dataset, do not converge to the same hypothesis function. Rather, the two hypotheses "disagree" in their predictions, at a rate roughly equal to the test error. This property seems at-odds with the convergence that's claimed in the present paper, and I'd be curious to hear the author's thoughts on this.
>
>  Thank you for sharing these relevant and interesting references. Disagreement in [2] compares two hypotheses, while Assumption 1 is about two (time-invariant and ergodic) *distributions* over weight/hypothesis space.  Assumption 1 considers a *distribution* of losses, by taking an average over time, i.e., over the trajectory of weights. It asks that the distribution of loss values along the trajectory converges (stationarity/ergodicity), this does not mean that the hypothesis converges to a fixed one (or, even that the loss converges to a fixed value).
>
> Since Assumption 1 is a statement about distributions, at individual time points, hypotheses can still disagree.
>
> In fact, Assumption 1 also applies to a distribution of (reasonable) initializations $w_0$ on $\mathbb{W}$; it means that after the loss function converges in distribution, the actual initialization is “forgotten”. In that case, the loss distribution still depends on the algorithmic parameters, dataset etc, but not on $w_0$ for almost every $w_0.$

---

### Official Review · Reviewer_3UBZ · 2022-07-11

**Rating:** 4
**Confidence:** 2
**Soundness:** 2 fair
**Presentation:** 1 poor
**Contribution:** 2 fair

**Summary:**

The standard notion o stability assumes learning algorithms to be convergent to a fixed point w.r.t. the weights. But, in practice, the weights of deep neural networks optimized with stochastic gradient descent do not satisfy such a condition and they often oscillate indefinitely. Motivated by this fact, the authors propose a notion of statistical algorithmic stability (SAS) that extends classical algorithmic stability to algorithms in which the weights only converge in distribution. The authors also show particular conditions under which such stability condition is satisfied and they discuss the learning algorithms from a dynamical systems perspective. Finally, they confirm their theoretical findings by computational experiments.

**Questions:**

In my opinion, the relation between the standard notion of stability and the SAS stability is not well investigated in the paper. Which is the relation between them? What about the generalization property of the two?

**Limitations:**

I do not see any negative societal impact.

**Strengths And Weaknesses:**

The problem addressed by the authors is for sure an interesting question. On the other hand, I think that the results could be presented in a clearer way, adding some examples and more discussion would make the paper for sure more readable by the community. For instance, adding specific examples of SAS algorithms would be very useful. Also the presentations of the theoretical results could be better addressed and the reader could be better guided in the meaning of the paper. For instance, giving a sketch of the proof of Thm.1 would be very meaningful. Otherwise, the paper remains quite abstract.

---

> ### Author Response · Authors · 2022-08-02
> **SAS and algorithmic stability are different concepts leading to different mechanisms for generalization; SAS applies to a larger class of algorithms that do not necessarily converge to a fixed point.**
>
> We thank the reviewer for their time and for suggesting improvements to our paper.
>
> > Which is the relation between them? For instance, adding specific examples of SAS algorithms would be very useful.
>
> Classical algorithmic stability is limited to algorithms that converge to a fixed point; it characterizes the robustness of the loss function at that fixed point to input perturbations. The new notion of SAS proposed in this paper is different from classical algorithmic stability. It is designed to apply to a class of learning algorithms that may or not converge to a fixed point. Thus, the robustness to perturbations that is analyzed in SAS is not of the loss function at a fixed point but rather of the ensemble statistics of the loss function. As explained in section 3 of the paper, the SAS coefficient, $\beta$ is a scalar function on the space of learning algorithms, while classical algorithmic stability coefficient is a scalar function on the space of weights/hypotheses.
>
> The idea of our two main results (Theorems 1 and 2) is to explain what an SAS algorithm is and what characterizes an SAS algorithm, respectively. Both these results apply to standard algorithms, such as gradient descent (GD), stochastic gradient descent (SGD), or variations (e.g., Adam). The generalization bound in Theorem 1 suggests that an algorithm (e.g. SGD, GD or any other optimizer such as ADAM) that has a higher value of $\beta$ is less SAS (stable) than an algorithm  with a lower value of $\beta.$ Theorem 2 indicates that between two optimizers, the one that has a lower value of $\beta$ (more stable) is the one whose statistics converge to their equilibrium values faster. This rate of convergence to equilibrium crystallizes the dependence of SAS on the algorithm and its parameters such as the learning rate as well as the loss function, the architecture, the dataset size and so on. As a result, all these factors affect the stability of a learning algorithm in the sense proposed in the paper. As a concrete example, (see Appendix section E for details), suppose SGD/GD at constant rate is SAS in the neural tangent kernel (NTK) regime. The same optimizer with the same learning rate becomes less SAS within the NTK regime when the dataset size is decreased, or when the dataset is perturbed such that the minimum NTK eigenvalue gets closer to zero.
>
> > Also the presentations of the theoretical results could be better addressed and the reader could be better guided in the meaning of the paper.
>
>  Thank you very much for this suggestion! We are revising some parts of the intro to make the big picture clearer and are revising the ends of section 4 and 5 to improve clarity of the theoretical results, their setting and the interpretation of the numerical results.
>
> > For instance, giving a sketch of the proof of Thm.1 would be very meaningful
>
> We have given a basic proof idea in the main paper, but will expand on it in the revision. The complete proof of Theorem 1 is available in Appendix B.
>
> > In my opinion, the relation between the standard notion of stability and the SAS stability is not well investigated in the paper. Which is the relation between them? What about the generalization property of the two?
>
>  In section 3.1, we discuss the difference between classical algorithmic stability and SAS. Mainly, SAS applies to a larger class of algorithms than classical stability: while classical stability assumes that the weight trajectories remain close, SAS does not make any such assumption, allowing for richer nonlinear dynamics (in the weight space) including limit cycles, quasiperiodicity and chaos. When the training dynamics converge to a fixed point, SAS reduces to standard algorithmic stability (as in Bousquet and Elisseef 2022). In other words, algorithmic stability is a special case of the more general SAS developed in this paper.
>
> Our main result in section 3.2 is the derivation of a generalization bound using SAS. We show that, when the training and generalization errors are defined appropriately, generalization bounds like those obtained using the classical notion of stability follow via SAS. This holds even for cases where the weights do not converge to a stationary point. In contrast, standard algorithmic stability does not apply to such cases.
>
> We will emphasize the differences more clearly in section 3 in our revision. Please also see our response to a related question above by reviewer jP3X. Thank you for the questions!

---

> > ### Comment · Reviewer_3UBZ · 2022-08-06
> > **Reply to the authors**
> >
> > Thanks for answering my questions.

---

> > > ### Author Response · Authors · 2022-08-08
> > > **Further questions we should clarify?**
> > >
> > > Thank you for reading our rebuttal.
> > >
> > > Are there any questions or concerns remaining that you would like us to address?

---

### Official Review · Reviewer_jP3X · 2022-07-15

**Rating:** 5
**Confidence:** 2
**Soundness:** 2 fair
**Presentation:** 2 fair
**Contribution:** 2 fair

**Summary:**

Traditional generalization analysis assume that a given algorithm converges to a fixed point. This paper studies the generalization of algorithms when the weights converge in distribution. In particular, the authors propose statistical algorithmic stability (SAS), through which they give upper bounds for the generalization bound. Then this paper gives a sufficient condition for an algorithm that satisfies the SAS.
Finally, this paper provides numerical results to find the SAS coefficient.

**Questions:**

- My main concern lies on the difference between the traditional algorithm stability and SAS in this paper.
The Definition 1 in this paper is based on a deterministic algorithm. However,  [1] gives a uniform stability definition with exception over the randomness of a random algorithm. That is to say, traditional algorithm stability can also handles the situations when learning algorithms that do not converge to a fixed point. I think it would be better if the authors could give more discussions on the comparison between the uniform stability in [1] and SAS.

- The generalization bound in Theorem 1 seems to be dominated by $\beta$. It would be more convincing if the authors could include numerical results on the order of $\beta$ when varying the sample size (on clean dataset rather than corrupted dataset)




[1] Hardt, Moritz, Ben Recht, and Yoram Singer. "Train faster, generalize better: Stability of stochastic gradient descent." International conference on machine learning. PMLR, 2016.

**Limitations:**

It seems that there is no potential negative societal impact of this work.

**Strengths And Weaknesses:**

Strengths:
- This paper is well organized and easy to follow.
- The idea of analyzing the generalization of algorithms that converge in distribution is interesting.
- Provides the code.


Weaknesses:
- The significance of SAS is not clearly stated when comparing with traditional algorithm stability.  By taking the expectation over the randomness of an algorithm, traditional algorithm stability can also analyzes the generalization when an algorithm does not converge to a fixed point. See the Questions part for details.


Minor issue:
- The references in the PDF cannot be retrieved. Maybe because the hyperref package is missing.

---

> ### Author Response · Authors · 2022-08-02
> **Statistical algorithmic stability is a different notion than algorithmic stability and applies to a larger class of learning algorithms**
>
> We thank the reviewer for asking a fundamental question about statistical algorithmic stability, the subject of our paper. In Hardt et al 2016 [1], which is referenced by the reviewer, the randomness in the definition of algorithmic stability (Definition 2.1) refers to extrinsic randomness associated with parameters of the algorithm. This is fundamentally different from the randomness on the weight space that we consider in our paper. We are editing section 3 in order to bring out the difference better. Below, we justify this difference in detail.
>
> 1. Definition 2.1 in [1] indeed takes an expectation with respect to randomness in the algorithm, which translates into a certain randomness in the weights. However, the proof in Section 3.1 in [1] relies on bounding, by a Lipschitz argument, the expectation $\mathbb{E}\|w_t-w_t'\|$ where $\mathbb{E}$ is an expectation over algorithmic randomness and $w_t ,w_t’$ are for two datasets differing in one point.  This means, in expectation over algorithmic randomness, weight trajectories diverge over time, and stability is ensured by limiting the number of steps (training fast). SAS, in contrast, allows large deviations $\|w_t – w'_t\|$, as long as the cumulative time average of the loss function converges, which is a weaker assumption. That is, training fast is not necessary for generalization via SAS, rather training algorithms for which long-term averages are robust generalize better.
> 2. SAS takes an expectation over time (i.e., the length of the trajectory), and then looks at the difference of the expectations for two datasets $S$ and $S'$. As a result, for any specific fixed $t$, $ \| w_t – w'_t \| $ can be large, as long as the expectations of the losses are close. For the same reason, SAS does not need to limit the step size.
> 3. In addition to the expectation over time, SAS can also account for additional randomness of the algorithm, e.g., SGD. We can either condition on the batch selection, or take an expectation over it (as in [1]).
>
> Thank you for the observation about $\beta$ dominating the generalization bound! In our section 4, Theorem 2, we derive an expression for the order of $\beta$. In particular, we show that under the assumptions outlined in section 4, $\beta \sim {\cal O}(1/n)$, where $n$ is the number of samples. However, our generalization bound (Theorem 1) admits an even slower decay of $\beta$ with $n,$ e..g, $\beta \sim {\cal O}(1/\sqrt{n}).$ These results suggest similar or faster decay rates of $\beta$ with sample size, when compared to the standard algorithmic stability coefficient (c.f. Feldman & Vondrak 2018)
>
> In our numerical experiments, we illustrate the central point of our theory that relates $\beta$ to with the dynamics of training (in particular, the rate at which statistics converge during training). Thus, we kept the training size constant throughout the experiments. We will work on numerical results for $\beta$ at different sample sizes and add them to an appendix, as you suggest.
>
>
> Thank you for pointing out the non-redirecting URLs! We have added the hyperref package in the revision.

---

> > ### Author Response · Authors · 2022-08-08
> > **Further questions we can clarify?**
> >
> > Thank you for reading our rebuttal.
> >
> > Are there any remaining questions or concerns that you would like us to address?

---

### Meta-Review · Area_Chair_d1QF · 2022-08-26

**Recommendation:** Accept
**Confidence:** Less certain

**Metareview:**

The authors new notation for the statistical algorithmic stability. It will be useful for studying algorithmic stability, and will open up new discussion in the field. Overall, the review team provide positive feedback and I would recommend accepting this work.


**Award:**

No

---

### Decision · Program_Chairs · 2022-09-14

Accept